# Trends in surface radiation and cloud radiative effect at four Swiss sites for the 1996 – 2015 period

Stephan Nyeki[1], Stefan Wacker[2], Christine Aebi[1,3*], Julian Gröbner[1], Giovanni Martucci[4], and Laurent Vuilleumier[4]

[1] Physikalisch-Meteorologisches Observatorium/World Radiation Center, Davos, Switzerland.
[2] Deutscher Wetterdienst, Meteorologisches Observatorium Lindenberg/Richard-Aßmann-Observatorium, Lindenberg, Germany.
[3] Oeschger Center for Climate Change Research and Institute of Applied Physics, University of Bern, Bern, Switzerland.
[4] Federal Office of Meteorology and Climatology MeteoSwiss, Payerne, Switzerland.
* Now at Royal Meteorological Institute of Belgium, Brussels, Belgium.

Correspondence to: Stephan Nyeki (stephan.nyeki@pmodwrc.ch)

**Abstract.** The trends of meteorological parameters and surface downward shortwave and longwave radiation (DSR, DLR) were analysed at four stations (between 370 and 3580 m asl) in Switzerland for the 1996 – 2015 period. Ground temperature, specific humidity and atmospheric integrated water vapour trends were positive during all-sky and cloud-free conditions. All-sky DSR and DLR trends were in the ranges 0.6 – 4.3 W m$^{-2}$/decade and 0.9 – 4.3 W m$^{-2}$/decade, respectively, while corresponding cloud-free trends were -2.9 – 3.3 W m$^{-2}$/decade and 2.9 – 5.4 W m$^{-2}$/decade. Most trends were significant at the 90 % and 95 % confidence levels. The cloud radiative effect (CRE) was determined using radiative transfer calculations for cloud-free DSR and an empirical scheme for cloud-free DLR. CRE decreased in magnitude by 0.9 – 3.1 W m$^{-2}$/decade (only one trend significant at 90 % confidence level), which implies a change in macrophysical and/or microphysical cloud properties. Between 10 and 70 % of the increase in DLR is explained by factors other than ground temperature and IWV. A more detailed, long-term quantification of cloud changes is crucial and will be possible in the future as cloud cameras have been measuring reliably at two of the four stations since 2013.

## 1 Introduction

Downward shortwave and longwave radiation (DSR, DLR) are important terms in the surface radiation budget and are fundamental in understanding the climate effect of increasing greenhouse gas concentrations (Wang and Dickinson, 2013). Both DSR and DLR have been reliably and accurately monitored since the late 1980s/early 1990s in several ground-based networks including: i) the Baseline Surface Radiation Network (BSRN) (König-Langlo et al., 2013; Driemel et al., 2018), ii) the Atmospheric Radiation Measurement (ARM) program (Ackerman and Stokes, 2003), and iii) the Surface Radiation (SURFRAD) Network (Augustine et al., 2000). DSR, from earlier less reliable measurements, over Europe was observed to decrease in the 1950s to 1980s ("dimming"), and was followed by an increase ("brightening") to the present, which has been attributed to changes in cloud cover and/or aerosol concentrations (e.g., Wild, 2009; Wang and Dickinson, 2013; Wild, 2016a; and references therein). DLR has also been observed to increase during the 1973 – 2008 period (Wang and Liang, 2009) and since the 1990s (Wild, 2016b), but the reliable observational record is in general only several decades long at present.

In support of these international efforts, the Alpine Surface Radiation Budget (ASRB) network was established in 1994/1995 at 11 stations in Switzerland to monitor regional radiation fluxes (Philipona et al., 1996; Marty, 2000; Marty et al., 2002). In a trend analysis of the 1995 – 2002 DLR time series at these stations, it was observed that average DLR increased by 5.2 and 4.2 W m$^{-2}$ for all-sky and cloud-free conditions, respectively (Philipona et al., 2004). A later study found an average cloud-free longwave increase of 3.5 W m$^{-2}$/decade for the 1996 – 2007 period at four of these stations (Wacker et al., 2011a), which were still in operation. It was estimated that >50 % of the DLR trend was due to the positive trends in temperature and

humidity. However, clouds can also significantly modify the radiation budget by reflecting shortwave and emitting longwave radiation. In order to quantify this with respect to the radiation budget, the concept of a Cloud Radiative Effect (CRE) can be used, which is the difference between all-sky radiation fluxes and cloud-free simulated fluxes (Ramanathan et al., 1989). Macrophysical (e.g., cloud cover, cloud base height, cloud top height, etc.) and/or microphysical cloud properties (e.g., cloud optical thickness, cloud droplet size, cloud particle size distribution, liquid water content, liquid water path, ice water content, hydrometeor size, hydrometeor size distribution, hydrometeor phase, etc.) can affect CRE to varying degrees. In a previous study at the same four Swiss stations, Wacker et al. (2013) determined that CRE increased by up to 7.5 W m$^{-2}$ over the 1996 – 2010 period, which was tentatively attributed to a reduction in the fractional cloud cover (FCC) or a change towards a different cloud type.

This study presents an update of radiation fluxes for the 1996 – 2015 period, spanning 20 continuous years of surface radiation measurements at each of the four Swiss stations. Our objectives are: i) to assess whether trends in all-sky and cloud-free surface radiation can be determined and explained with any greater certainty, ii) to assess the trends in the shortwave, longwave and total CRE, and iii) apply a wider range of robust statistical techniques than in previous studies.

## 2 Methods

### 2.1 Data from ASRB and SACRaM Networks

The ASRB network monitored DSR and DLR at 11 existing stations belonging to the Swiss Federal Institute of Meteorology (MeteoSwiss) from 1994 to 2005. Measurements were conducted according to BSRN guidelines, published in a 2005 report by McArthur (2005). In a subsequent rationalization of the network, only four of the original eight stations continued to operate. The remaining stations are (in order of altitude): Locarno (LOC, 46.180°N, 8.783°E, 367 m), Payerne, (PAY, 46.815°N, 6.944°E, 491 m), Davos (DAV, 46.814°N, 9.846°E, 1594 m), and Jungfraujoch (JFJ, 46.549°N, 7.986°E, 3580 m). Instruments from these stations were incorporated into the MeteoSwiss CHARM (Swiss Atmospheric Radiation Monitoring) network, which were then progressively merged in 2007 – 2012 into a single network, the Swiss Alpine Climate and Radiation Monitoring network (SACRaM). Several aspects concerning the instruments are worth mentioning and are therefore briefly discussed here:

i) Pyrgeometers in the ASRB network were all unshaded, and hence a correction for solar heating of the instrument was applied using the method described by Dürr (2004). In contrast, it was unnecessary to correct DLR data from the SACRaM network. These were either shaded (Precision Infrared Radiometer, PIR, Eppley Inc., USA) or unshaded pyrgeometers (CG(R)4, Kipp & Zonen, Netherlands). As CG(R)4 pyrgeometers are less affected by heating effects or by longwave irradiance in the direct beam of the sun (Meloni et al., 2012; Gröbner et al., 2018), no correction is necessary.

ii) The SACRaM data acquisition systems were updated in stages from March 2005 to October 2011, which resulted in several short monitoring gaps. For instance, monitoring at PAY was interrupted from 23 August 2011 to 1 November 2011, but was not considered to be long enough to affect the trend analysis in this study.

iii) SACRaM radiometers at DAV are located at and maintained by the Physikalisch-Meteorologisches Observatorium Davos/World Radiation Center (PMOD/WRC). Due to major building renovation from December 2010 to September 2012, these radiometers were partially removed from December 2010 to December 2014, however, PMOD/WRC radiometers were re-located nearby. DSR data from the SACRaM network was available while DLR data from the World Infrared Standard Group (WISG) of pyrgeometers (WMO, 2006) was used for the January 2006 to December 2015 period instead. The WISG consists of four pyrgeometers, which were averaged into a single DLR time series of 1-min data.

iv) The PMOD/WRC hosts the World Standard Group (WSG) of pyrheliometers and the WISG, as mentioned above. These provide the reference scales for shortwave and longwave radiation measurements, respectively. However, as a note of interest, several studies have determined that their reference scales may need to be revised in the future (Fehlmann et al., 2012; Gröbner et al., 2014). The WSG scale currently overestimates by +0.3 %, and a linear correction could be applied in a straightforward manner. However, the WISG scale underestimates longwave fluxes, which will require a non-linear correction depending on a number of factors (e.g., raw signal data, etc.), as reported by Gröbner et al. (2014) and Nyeki et al. (2017). The latter study determined that corrections were in the ranges 1 to 4 W m$^{-2}$ for all-sky DLR and 5 to 7 W m$^{-2}$ for cloud-free DLR when based on available data from three BSRN stations and Davos (i.e., PMOD/WRC), which have the longest time series. Such corrections are beyond the scope of the present study, and are currently being debated within the community. A possible future correction of the SACRaM DSR time series should have no effect on the trend analyses in this study while corrected DLR time series could marginally affect the trends depending on the degree of cloudiness at each station.

v) The uncertainty of pyranometer measurements is estimated to be in the range 18 – 23 W m$^{-2}$ for high intensity (1000 W m$^{-2}$) 1-min average values (Vuilleumier et al., 2014). Similarly, the uncertainty of pyrgeometer measurements is estimated at ±4 W m$^{-2}$, and their relative stability is within ±1 W m$^{-2}$ over extended time periods (Gröbner et al., 2014; Nyeki et al., 2017).

vi) Meteorological data (10-min. resolution) were available as quality controlled and assured data from MeteoSwiss: including screen-level temperature 2 m above ground ($T_{2m}$), relative humidity (RH), and surface pressure ($P_s$). In addition, DSR and DLR data (1-min.) were also available. Integrated water vapour (IWV; 1-hour) from Global Navigation Satellite System (GNSS) measurements (Morland et al., 2006) was downloaded from the STARTWAVE database (www.startwave.ch). The specific humidity (SH) was calculated using $T_{2m}$, RH and $P_s$. IWV measurements from JFJ were not used for a number of reasons as previous studies (Nyeki et al., 2005; Morland et al., 2006) had concluded that GNSS IWV time series at JFJ were uncertain due to: i) a high variability in IWV values, and ii) the IWV retrieval algorithm was unable to adequately correct for the persistent influence of snow and ice on the GNSS antenna signal. As a result, IWV at JFJ was based on a commonly-used parameterisation by Leckner (1978) using $T_{2m}$ and RH. Gubler et al. (2012) estimated that the uncertainty in IWV using this parameterisation was up to 100 %.

Monthly average values were then constructed for time series analysis. Use of a method by Roesch et al. (2011) was considered, which minimises the risk of biased monthly mean values when calculated from incomplete or flagged data records of DSR and DLR. A comparison of results for all sky conditions with simple monthly averages gave results, which were different by <0.1 %. Hence, for the sake of consistency and comparability, simple monthly averages were used throughout this study for the trend analyses. A monthly average was accepted for all-sky conditions if ≥75 % of data were available for each month, while no sampling threshold was applied to cloud-free data due to the smaller dataset after application of the cloud filter.

## 2.2 Determination of Cloud-Free Conditions

In order to calculate cloud-free climatologies of meteorological parameters and radiation fluxes, it was necessary to determine the occurrence of cloud-free conditions. The first method uses $T_{2m}$, RH and DLR as input data to a semi-empirical algorithm, the Automatic Partial Cloud Amount Detection Algorithm, APCADA, (Dürr and Philipona, 2004). The degree of cloudiness, can be derived in oktas (0 to 8) and then converted to FCC (1 okta = 0.125 FCC) for any 10-min period during any time of the day. Cloud-free versus cloudy cases can be distinguished with an uncertainty of about 5 % for low to mid-level clouds. APCADA has the advantage that night-time FCC data can be derived for the four locations in this study based on previous semi-empirical studies (e.g., Dürr and Philipona, 2004). However, APCADA has several minor drawbacks. The first is a

difficulty in adequately detecting high-altitude clouds (particularly optically thin cirrus) because of their low radiative impact at the surface. Nevertheless, as the radiative effect of such clouds on DLR is small, the effect of cloud contamination in the cloud-free dataset is also considered to be small. The second drawback is that APCADA semi-empirical calibration values (lapse rate coefficient and effective cloud-free broadband emissivity) are based on climatological conditions at each location

in the early 1990s. While these calibration values are not expected to have changed since then, this cannot be verified here without an updated analysis. An alternative method, presented by Long and Turner (2008), determines the cloud cover using meteorological parameters and various statistical thresholds based on current data. It was argued that cloud-free estimates were more accurate, but a comparison with APCADA remains to be conducted in a future study. Apart from these aspects, the use of proxy parameterisations for cloud cover will introduce uncertainties, but we estimate that these are generally low. A more

accurate assessment will only be possible when cloud cover data from sky cameras are long enough to conduct reliable time series analysis, which is generally a period of 10 years and longer. While cloud cover can be accurately and objectively determined with sky cameras, measurements are only available during daylight hours. Sky cameras were installed in 2013 at PAY (VIS-J1006, Schreder GmbH) and DAV (Q24M, Mobotix), while several difficulties at JFJ have prevented reliable measurements. Hence, continuous FCC time series, with a length of about six years are only available at two stations. However,

we used sky camera data to assess whether improvements could be made to the APCADA method. Images taken at PAY have a temporal resolution of five minutes, and two are sequentially taken with different exposure times (1/500 and 1/1600 s) having a resolution of 1200 x 1600. One image is taken each minute at Davos with an exposure time of 1/500 s. After the pre-processing of images (Aebi et al., 2017), a colour ratio (the sum of the blue to green ratio plus the blue to red ratio) is calculated per pixel (Wacker et al., 2015) and compared to empirically determined reference values (2.2 in DAV and 2.5 in PAY), which

are based on a large database of sky camera images. A pixel is classified as being cloudy or cloud-free based on this comparison. The FCC is then calculated by summing up the cloudy pixels and dividing by the total number of pixels. FCC values ≤0.05 for each 10 min value were categorised as cloud-free conditions, which is more stringent than for APCADA where the limit is ≤ 1 okta (i.e., FCC ≤ 0.125).

**2.3 Parameterisation of Cloud-Free DSR and DLR**

As mentioned in Section 1, the effect of clouds on the surface radiation budget can be expressed by the CRE (Eq. 1), which is divided into components for the shortwave and longwave cloud effects (SCE and LCE, respectively). Each component itself is defined as the difference between all-sky fluxes (e.g., $DSR_{all\text{-}sky}$) and corresponding simulated cloud-free fluxes (e.g., $DSR_{sim\,cloud\text{-}free}$), as in Eq. 2:

$$CRE = SCE + LCE, \tag{1}$$

$$CRE = DSR_{all\text{-}sky} - DSR_{sim\,cloud\text{-}free} + DLR_{all\text{-}sky} - DLR_{sim\,cloud\text{-}free}, \tag{2}$$

CRE is defined here using just the downward flux components, similar to other studies (e.g., McFarlane et al., 2012), rather

than the net (i.e., downward – upward) fluxes (e.g., Berg et al., 2011), so care must be taken when comparisons are made. $DSR_{sim\,cloud\text{-}free}$ in Eq. 2 was calculated using the solar zenith angle, IWV, and aerosol optical depth (AOD) as inputs to libRadtran (Library for Radiative Transfer) (Mayer and Kylling, 2005). AOD from sun-photometers at each of the four sites was derived using procedures and data published previously (Nyeki et al., 2012; Kazadzis et al., 2018). AOD data (1 min.) was only available for Jan. 1994 – Dec. 2012, which was used to construct an AOD climatology for the Jan. 2013 – Dec. 2015

missing period. While this may introduce an error in the AOD trend, a large change is not expected as the measured time series is 18 years long.

$DLR_{sim\,cloud-free}$ was calculated using the empirical parameterisation by Prata (1996) as in Eq. 3:

$$DLR_{sim\,cloud-free} = (1 - (1 + w).exp(-(1.2 + 3.w)^{0.5})). \sigma T^4_{2m}, \tag{3}$$

Where $T_{2m}$ is in Kelvin, $\sigma$ is the Stefan-Boltzmann constant ($5.67 \times 10^{-8}$ W m$^{-2}$ K$^{-4}$), $e$ is the water vapour pressure (hPa), and $w = 46.5e/T_{2m}$. As $w$ is in fact the parameterisation for IWV (in cm), observed values of IWV from GNSS measurements were used instead. A slightly modified form of the above Prata parameterisation was developed by Gröbner et al. (2009) by using the effective atmospheric boundary layer temperature ($T_{ABL}$) instead of $T_{2m}$. $T_{ABL}$ represents the effective radiating temperature of water vapour in the atmospheric boundary layer, and is derived by using two co-located pyrgeometers: one standard

pyrgeometer sensitive to the $3 – 50\,\mu$m wavelength range and another modified one, which is sensitive in the $8 – 14\,\mu$m range. Setting $T_{2m} = T_{ABL}$, as well as use of the Prata parameterisation was considered by Wacker et al. (2014) to be slightly more accurate than the modified Brutsaert parameterisation used by Wacker et al. (2011a). The former was therefore used as the main parameterisation of $DLR_{sim\,cloud-free}$ in this study.

       Validation of the cloud-free models was accomplished by determining the shortwave and longwave discrepancies

(observed cloud-free fluxes – simulated cloud-free fluxes). The mean bias and rmse of the shortwave discrepancies were <3.5% and <8.5% (Wacker et al., 2013), respectively, and ~ -0.1 W m$^{-2}$ and ~3.9 W m$^{-2}$ for the longwave discrepancies at all four stations. The mean biases are thus similar to the measurement uncertainty of the respective radiometers (Wacker et al., 2013).

       An alternative parameterisation of $DLR_{sim\,cloud-free}$, reported by Ruckstuhl et al. (2007), was briefly investigated as

well. Using data from the same four Swiss SACRaM stations, Ruckstuhl et al. (2007) parameterised $DLR_{sim\,cloud-free}$ using only GNSS-derived IWV and not $T_{2m}$. A power-law of the following form was found for this DLR-IWV parameterisation when data from all four stations was combined into a single equation:

$$DLR = a.IWV^b, \tag{4}$$


where the coefficients $a$ and $b$ were calculated for cloud-free conditions. It was determined that observed and parameterised monthly values for the 2001 – 2004 period gave correlation coefficients $R^2$ >0.95 and had root-mean-square errors (rmse) of 9.2 – 12.0 W m$^{-2}$. Ruckstuhl et al. (2007) concluded that $DLR_{sim\,cloud-free}$ could be parameterised with an uncertainty of <5 % when based on monthly average values. The main reason for including this method here, is to test whether $DLR_{sim\,cloud-free}$ can

be even more accurately parameterised with longer IWV time series in order to calculate LCE. This method was also used to test Eq. 4 during all-sky and not just cloud-free conditions.

**2.4 Statistical Methods**

Trend analyses were performed using several methods. The first was the linear least squares (LLS) method by Weatherhead et al. (1998), using de-seasonalised monthly average values. Further details are given in the prior study by Wacker et al. (2011a).

The second method uses the seasonal Kendall test and Sen's slope estimator (see Gilbert, 1987; and references therein). The seasonal Kendall test is an extension of the Mann-Kendall test, a non-parametric technique, which determines whether a monotonic positive or negative trend exists. The test takes seasonal effects into account and hence avoids the problem of auto-correlation in the time series. Before these trend tests were applied, the homogeneity of the time series was checked using three tests: the Buishand test (parametric), the Pettitt test (non-parametric), and the standard normal homogeneity test (SNHT;

parametric) (Wijngaard et al., 2003). The null hypothesis is that the time series is homogeneous (significance level $p > 0.05$), while a stepwise change in the mean (or other statistic) is present under the alternative hypothesis ($p < 0.05$). When correctly

used, these tests can help to locate when a possible change occurred. The SNHT test is more sensitive to changes near the beginning and end of a time series, whereas the Buishand and the Pettitt tests are more sensitive to changes in the middle. In order to meet the normality assumption for the SNHT and Buishand tests, monthly time series were log-transformed.

## 3 Results and Discussion

### 3.1 Meteorological and Surface Radiation Climatologies

Tables 1 and 2 summarise meteorological and radiation flux statistics for all-sky and cloud-free climatologies, respectively, at all four stations. Seasonal averages (DJF, MAM, etc.) clearly illustrate an annual cycle in virtually all parameters with a maximum in summer and minimum in winter. All-sky values in Table 1 have not been previously reported while cloud-free values in Table 2 are similar to values reported by Wacker et al. (2011a) for the 1996 – 2007 period. $T_{2m}$ and DSR values are seen to be slightly lower in Table 1, as would be expected during cloudy conditions. In contrast, SH and IWV are higher during all-sky conditions, which in turn results in higher DLR values.

Table 1 also shows cloudiness at each station from APCADA results, which have been converted from oktas to FCC. The clearest conditions occur at Locarno (lee location, south of the Alps) with an average FCC value = 0.55 while the cloudiest conditions occur at PAY (plateau location, north of the Alps) with FCC = 0.70 as a result of more persistent stratus cloud cover particularly during winter time when low cloud type stratus nebulosus regularly covers the Swiss Plateau.

To demonstrate the annual cycles in surface radiation at all four stations, DLR time series for all-sky and cloud-free conditions are shown in Figure 1a-d. Maxima in summer and minima in winter are evident as also is the case for DSR (not shown). Lower annual average DLR values during cloud-free conditions are observed with increasing station altitude (Table 2: 289 W m$^{-2}$ at LOC versus 175 W m$^{-2}$ at JFJ) as reported by Marty et al. (2002) for the same stations. This generally occurs as a result of lower IWV and temperature values with increasing altitude but is not always strictly the case as each station has its own climatology. For instance, the average DLR at PAY in Table 2 is very similar to that at LOC despite the latter being 124 m lower in altitude. When considering average DSR values with altitude, the situation is similar during cloud-free conditions except that higher long-term averages are generally observed with increasing altitude due to the decrease in atmospheric optical depth (Marty et al., 2002; and references therein). Again, the climatology at each station also has an influence as the cloud-free annual average DSR at LOC (229.3 W m$^{-2}$) is higher than at PAY (206.8 W m$^{-2}$) and DAV (216.2 W m$^{-2}$).

### 3.2 Meteorological and Surface Radiation Trends

A summary of the decadal trends (LLS and Sen's slope methods) of all parameters is shown in Table 3. Trend values and confidence levels for both methods are seen to closely agree (i.e., column 3 vs. 4, and 5 vs. 6) in most cases, which gives confidence in their use. However, apparent discrepancies may occur on occasion when time series consist of many outliers or trends are close to zero. In these cases (e.g., IWV at PAY during all-sky conditions in Table 3) results from the Sen's slope method are preferred as they are considered to be more robust to outliers than the LLS method as well as being more accurate when data are skewed (Wilcox, 2005). In order to be consistent with prior studies (Wacker et al., 2011a; 2013), results from the LLS method will mainly be discussed here unless otherwise stated. The 90 % confidence interval of each trend is also shown in Table 3. Intervals are only shown for the LLS method for clarity.

Trends in $T_{2m}$, SH and IWV, in Table 3, are all positive during all-sky and cloud-free conditions. More specifically, $T_{2m}$, SH and IWV have increased at all four stations during all-sky and cloud-free conditions on average by ~0.3 – 0.6°C/decade, ~0.1 – 0.2 g kg$^{-1}$/decade and 0.2 – 0.8 mm/decade, respectively. It is interesting to note that about three quarters of the all-sky and cloud-free trends in meteorological parameters are significant at the >90 % confidence level. Homogeneity analyses of all meteorological parameters were then conducted to test for any discontinuities in the time series. This is only

meaningful when using the full dataset i.e., for all-sky conditions as opposed to cloud-free conditions, which are a sub-set of the former. Results from the SNHT, Buishand and Pettitt homogeneity tests indicate that no time series at any station had $p <$ 0.05, suggesting that all meteorological time series were homogeneous with no significant discontinuities due to climatic or non-climatic effects such as a change of instrument or data acquisition system, relocation, etc.

5   Trends in all-sky DSR are in the 0.6 – 4.3 W m$^{-2}$/decade range, and are significant at the 90 % confidence level except for DAV. Cloud-free trends for DAV and LOC are similar at 3.1 and 3.3 W m$^{-2}$/decade, respectively, but are rather different for PAY and JFJ. On closer inspection, the DSR trend at PAY is not monotonic but exhibits a trend of 2.9 W m$^{-2}$/decade for 1996 – December 2011 followed by a more positive trend, resulting in an overall trend of 10.6 W m$^{-2}$/decade for 1996 – 2015. A similar case occurs at JFJ, where the trend for 1996 – Dec. 2007 is -2.9 W m$^{-2}$/decade followed by a more negative trend, resulting in an overall trend of -9.5 W m$^{-2}$/decade for 1996 – 2015. Only the Pettitt homogeneity test suggested that a discontinuity in the DSR trend occurred at PAY and JFJ (both, $p < 0.05$). No discontinuities were found for DAV or LOC DSR trends. At present, the reason(s) for these cloud-free trends at PAY and JFJ for 1996 – 2015 are unknown and will have to be further monitored. The SCE, LCE and CRE are not affected by these results as they are calculated with all-sky data.

   Regarding the DLR trends, all are positive and significant at the >90 % confidence level except during all-sky conditions at PAY. All-sky DLR trends at the four stations range from 0.9 – 4.3 and 0.9 – 5.9 W m$^{-2}$/decade for the LLS and Sen's methods, respectively. Higher trends are found for cloud-free conditions with ranges from 2.4 – 5.4 and 2.5 – 5.9 W m$^{-2}$/decade, respectively, while all trends are significant at the 95 % confidence level. The magnitudes and direction of the trends are similar to those observed by Wacker et al. (2011a; 2013) with the important exception that DLR time series trends are now significant for virtually all cases (i.e., combinations of stations, cloud conditions, and statistical tests), which was previously observed for only two cases.

   We found stronger cloud-free DLR trends at mountain stations (DAV and JFJ) than at lowland stations (LOC and PAY). This seems to be in agreement with a review by Pepin et al. (2015), who claim that climate warming is stronger at higher elevations, an effect known as elevation-dependent warming. However, in our study the cloud-free temperature trends are actually smaller at mountain than at lowland stations. This could be related to the temperature trends including the combined effect of multiple factors depending on local climate conditions, such as cloudiness. On the other hand, the cloud-free DLR trends are more closely linked to the driver of climate change: the increasing absorption of the upward longwave flux by the atmosphere and subsequent reemission in all directions including DLR. Pepin et al. (2015) formulated several hypotheses to explain their findings. The one that seems most consistent with our findings postulates that an increase in DLR is related to an increase in IWV since we found stronger IWV changes at mountain stations in relative terms. At DAV, the cloud-free IWV trend is larger than at the lowland station, even though the average IWV is significantly smaller. At JFJ, the cloud-free IWV trend is smaller than at the lowland station, by a factor up to two, but the average IWV is almost four times smaller than at the lowland stations. However, as will be shown in section 3.3.2, changes in $T_{2m}$ and IWV are not sufficient to explain the change in cloud-free DLR at mountain stations.

   The 90 % confidence intervals of the DLR trends, as well as those for meteorological and DSR trends, are shown in Table 3. Interval values are relatively low in all cases, and is in large part due to the long time series. If the instrumental uncertainties are taken into account by the trend analysis, then 90 % confidence intervals are unchanged to two decimal places. However, our main reason to have confidence in trend results, rests on whether they are significant or not at the 95 % confidence level, which has been demonstrated in Table 3.

   How do DSR and DLR trends at the four Swiss stations compare to other regions or global averages? In a recent analysis of observed DSR trends at BSRN stations, Wild (2016b) found an overall increase of 2.0 W m$^{-2}$/decade since the 1990s during all-sky conditions and a similar value during cloud-free conditions. The study concluded that a reduction in aerosol concentrations was contributing to the increase in DSR. Studies of trends in DLR are scarcer. Apart from the earlier mentioned studies (Philipona et al., 2004; Wacker et al., 2011a; 2013), which focused on the ASRB network in Switzerland, a

global increase of 2.2 W m$^{-2}$/decade in DLR was estimated for the 1973 – 2008 period (Wang and Liang, 2009) using temperature, humidity and cloud fraction to parameterise DLR. A lower trend of 1.5 W m$^{-2}$/decade was found in climate model simulations of the Coupled Model Intercomparison Project Phase 5 (CMIP5) by Ma et al. (2014) for the 1979 – 2005 period. In a more recent study by Wild (2016b), 20 of the longest BSRN all-sky DLR time series had an overall average trend of 2.0

W m$^{-2}$/decade (11 significant) while three were negative (none significant). This agreed well with CMIP5 multi-model mean trends for two RCP scenarios (Representative Concentration Pathways 8.5 and 4.5), which gave all-sky trends of 1.7 and 2.2 W m$^{-2}$/decade, respectively.

### 3.3 SCE, LCE and CRE

#### 3.3.1 Trend Analysis

Time series of SCE, LCE and CRE, calculated according to Eqs. 1 and 2, are shown for PAY as an example in Figure 2. Beginning with a discussion of the SCE, all annual averages (see Table 4) are found to be negative with the lowest values (< -70 W m$^{-2}$) occurring at DAV and PAY. This can partly be explained by a higher cloud frequency at these sites with FCC = 0.68 and 0.70, respectively, agreeing with short-term results by Aebi et al. (2017). Positive trends of 3.6 and 3.8 W m$^{-2}$/decade (see Table 5) are observed at LOC and PAY, respectively, which represent a decrease in the magnitude of the SCE. In contrast,

SCE trends at DAV and JFJ are close to zero for both the LLS and Sen's slope methods. Neither LOC nor PAY trends are significant at the 95 % confidence level but their positive values arise from the fact that trends in DSR$_{\text{all-sky}}$ > DSR$_{\text{sim cloud-free}}$. Apart from DSR, DSR$_{\text{sim cloud-free}}$ is also calculated using IWV and AOD. IWV Trends at LOC and PAY in Table 3 are slightly positive but not significant. AOD trends at LOC and PAY are shown in Figure 3, and were calculated for 1996 – 2013 to be consistent with the 1996 – 2015 period used in this study. Both trends are negligible at 0.03 and 0.00/decade, respectively, and

remain the same if the full AOD time series from 1994 to 2013 is used instead. Decadal trends at DAV and JFJ are similarly negligible, as shown in a previous study (Nyeki et al., 2012) and in unpublished data. Positive SCE trends at LOC and PAY are therefore mainly due to positive trends in DSR$_{\text{all-sky}}$.

     Regarding the LCE, annual average values are all positive with the highest occurring at JFJ (49.9 W m$^{-2}$) and the lowest at LOC (23.3 W m$^{-2}$). LCE decreases with decreasing altitude due to the higher water vapour content and thus higher

cloud-free longwave fluxes (e.g., Wacker et al., 2011b; Aebi et al., 2017). LCE trends are negative at PAY and LOC, which are consistent with a decrease in the magnitude of the SCE and the lower all-sky DLR trends with respect to the cloud-free trends at these sites. In contrast, LCE trends at DAV and JFJ are positive, at 1.0 and 2.4 W m$^{-2}$/decade, but none are significant. The LCE depends on a range of microphysical and macrophysical cloud properties, as mentioned in Section 1. In a case study, Aebi et al. (2017) observed that low-level clouds (for example cumulonimbus-nimbostratus or stratus-altostratus) and a cloud

coverage of 8 oktas have the highest impact on the magnitude of the LCE with values of 59 – 72 W m$^{-2}$. The lower the cloud base height, the higher the cloud base temperature and the larger the LCE.

     As CRE is the sum of SCE and LCE, annual average values in Table 4 are more influenced by SCE than LCE, and result in DAV and PAY having the lowest values at ~ -40 W m$^{-2}$. It can be useful to regard results in Table 4 as representing a regional value for Switzerland when averaged over all four stations. The regional values of SCE, LCE and CRE are then -

61.6, 34.1 and -27.6 W m$^{-2}$, respectively. Interestingly, these values are similar to recently updated global average values of - 56, 28 and -28 W m$^{-2}$ reported by Wild et al. (2017) using BSRN observational data. The similarity is reflected by the fact that these globally averaged values are predominantly weighted by European as well as global mid-latitude sites with similar cloud climatologies. Regarding the CRE trends in Table 5, all are positive and range from 0.9 – 3.1 W m$^{-2}$/decade, which is similar to a range of 1.3 – 7.4 W m$^{-2}$/decade for 1996 – 2011 reported by Wacker et al. (2013). However, it should be noted that no

trends are significant at the 95 % confidence level with only PAY significant at the 90 % level. Although the absence of any significant trend hampers further reliable interpretation, it is nevertheless interesting to consider the possible meaning of results in Table 5. The positive values of the CRE trends represent an overall decrease in the CRE magnitude, and imply that changes

in macrophysical and/or microphysical cloud properties have occurred during the 1996 – 2015 period. Despite the observed decrease in CRE, clouds continue to reduce the available radiative energy at the surface over the four SACRaM sites by an overall long-term average of ~ -28 W m$^{-2}$. A decrease in cloud cover might be one of the cloud parameters, which has contributed to a decrease of the CRE magnitude. Indeed, a reduction in cloud cover over Europe and adjoining regions has been ascertained in several studies based on observations and simulations. Sanchez-Lorenzo et al. (2017) reported the observed and simulated cloud cover for the 1971 – 2005 period, and found negative trends during the first two decades over the Mediterranean region, followed by a subsequent tailing off. The region of study (30° – 48°N) also covered Switzerland (~46.2° – 47.6°N), which exhibited a weak, overall negative trend in cloud cover. It was argued that the northward expansion of the Hadley cell may be related to the observed changes in cloud cover over the Mediterranean region. In a further recent study based on satellite and BSRN data covering the 1983 – 2015 period, it was concluded that the major part of the overall positive trend in surface solar radiation over Europe was possibly due to changes in clouds (Pfeifroth et al., 2018). These aspects were more closely investigated with respect to possible changes in synoptic weather patterns by Parding et al. (2016). They observed that an increase in cyclonic and decrease in anticyclonic weather patterns occurred over northern Europe, and contributed to dimming in the 1960s to 1990s based on observational data from the Global Energy Balance Archive (GEBA; Gilgen and Ohmura, 1999).

Apart from changes in cloud cover and other macrophysical cloud properties, microphysical cloud properties can also have a substantial impact on CRE. However, the observation of these properties using active remote sensing techniques is limited to a few super-sites worldwide while long-term time series are rarely available. The same is also valid at the four Swiss SACRaM stations in this study. Selected macrophysical and microphysical cloud properties have only been routinely measured at PAY since 2010 and 2005, respectively. Cloud observations from human observers were unfortunately discontinued in 2000 – 2005 at all SACRaM locations.

### 3.3.2 Trends of the Longwave Discrepancies

Through analysis of the trends in the longwave discrepancy, it is possible to assess the strength of radiative forcing components other than due to changes in $T_{2m}$ and IWV. Only these latter two parameters are used in the Prata parameterisation to estimate DLR$_{sim\ cloud-free}$. Trend analyses of the longwave discrepancies are shown in Table 6 for all stations. The LLS DAV trend of 3.4 W m$^{-2}$/decade represents 70 % of the overall DLR trend of 4.8 W m$^{-2}$/decade from Table 3. A similarly high value is also found at JFJ, and suggests that 70 % of the overall cloud-free trends at these stations are due to factors other than $T_{2m}$ and IWV. In contrast, the LLS trend at LOC (10 % value) is almost fully explained by increases in $T_{2m}$ and IWV while PAY (51 % value) is partially explained.

Previous studies (Philipona et al., 2005; Wacker et al., 2011a) have investigated the trends in the longwave discrepancy using similar methods to those in this study. Possible changes in atmospheric gases or aerosol concentrations were investigated but not considered to substantially contribute to the discrepancy. It was noted that the increase in atmospheric $CO_2$ was responsible for a DLR trend of only ~0.3 W m$^{-2}$/decade (Prata, 2008) while increases in atmospheric $CH_4$ and $N_2O$ (Forster et al., 2007) resulted in a trend of ~0.01 W m$^{-2}$/decade. Furthermore, the effect of aerosols was assumed to be insignificant (Ramanathan et al., 2001). However, it was argued that the use of APCADA, to generate a cloud-free filter, was possibly a biasing factor. As mentioned previously, high-altitude clouds (e.g., cirrus) have a smaller effect on DLR than low or mid-altitude clouds, and hence the cloud-cover filter generated with APCADA may not be accurate during such conditions. If this is the case then a positive trend of the longwave discrepancy suggests an increase of the radiative effect of high-level clouds, whereas a negative trend indicates a decrease. Under such assumptions, the positive trends in Table 6 would therefore point to an increase of the radiative effect of high-altitude clouds over the 1996 – 2015 period.

Another interesting aspect in Table 6 is the apparently stronger trend (both LLS and Sen's methods) with increasing station altitude, although only trends at DAV and JFJ are significant. As mentioned in Section 3.2, the cloud-free DLR trends

themselves are higher at DAV and JFJ than at the other stations, which can be put into perspective with the findings of Pepin et al. (2015) and their hypothesis that DLR exhibits higher sensitivity to an IWV increase at low IWV, which is typically the case at mountain stations. The Prata parameterisation (Eq. 3) used in our study is a nonlinear function of IWV, but it seems to only partially explain the stronger cloud-free DLR trends at DAV and JFJ. It is possible that the higher uncertainty of IWV at JFJ plays a role or that the Prata parameterisation does not fully capture the DLR IWV dependency at mountain stations. Pepin et al. (2015) also discussed whether a change in the snowline altitude and subsequently surface albedo could occur. A further study based on radiative transfer modelling would be required to test these aspects.

### 3.3.3 Improvement of Methods

As mentioned earlier in Section 2.3, the DLR-IWV parameterisation is a possible alternative method to determine $DLR_{sim\,cloud\text{-}free}$. Figure 4 shows monthly average values of observed DLR versus IWV during all-sky and cloud-free conditions for the 2000 – 2015 period. DLR is seen to be less sensitive to changes in IWV at higher IWV values, which is due to saturation of longwave absorption in the atmospheric longwave window. The power-law fits (blue curves) in both graphs have been calculated for ≥684 monthly average values, and agree well with superimposed curves (red) from Ruckstuhl et al. (2007) for the 2001 – 2004 period. Fits for all-sky and cloud-free conditions exhibit values $R^2$ = 0.95 and 0.97 with rmse = 10.0 and 10.1 W m$^{-2}$, respectively. Despite the good overall fit, Figure 4 shows that the agreement becomes poorer when IWV ≲ 5 mm, especially during cloud-free conditions. These are mainly JFJ data points, which have a high uncertainty due to the aspects discussed earlier in section 2.1. The greater scatter in both graphs with respect to that of the modified Prata parameterisation (rmse < 4.0 W m$^{-2}$, all stations) therefore suggests that this straightforward parameterisation of cloud-free DLR using only IWV will not allow sufficiently accurate LCE trends to be determined, even with longer time series.

A promising alternative to APCADA to determine the degree of cloud cover is the use of sky cameras. However, FCC time series at DAV and PAY from sky camera data are only available as of 2013, and hence cannot be used to replace APCADA in this study based on the 1996 – 2015 period. Instead, the 2013 – 2015 FCC time series was tested with the above DLR-IWV parameterisation. Rmse values of 13.7 W m$^{-2}$ and 12.8 W m$^{-2}$ for all-sky (R$^2$ = 0.80) and cloud-free conditions (R$^2$ = 0.85) were obtained, respectively. These rmse values are higher than with APCADA (10.0 and 10.1 W m$^{-2}$) but are likely to improve (i.e., decrease) when longer FCC time series from sky cameras become available in the future. A further major refinement is the use of an infrared sky camera, which allows cloud cover to be determined during the day and night. A research prototype, the thermal infrared cloud camera (IRCCAM), has been continuously operating at DAV since September 2015 (Aebi et al., 2018). A comparison of IRCCAM with the visible sky cameras gave FCC values to within ±0.07 and to within ±0.05 for APCADA. Aebi et al. (2018) concluded that the use of FCC from infrared sky cameras could increase the accuracy of cloud-free climatologies when FCC time series of adequate length become available.

### 4 Conclusions

The trends of surface downward shortwave and longwave radiation (DSR, DLR) were analysed at four stations (between 370 and 3580 m asl) in Switzerland for the 1996 – 2015 period. Using these data and meteorological parameters, the cloud radiative effect (CRE) was determined from calculations of the shortwave and longwave cloud radiative effects. The main conclusions include the following:

1) Trends in $T_{2m}$, SH and IWV all increased during all-sky and cloud-free conditions. Two thirds were significant at the ≥90 % confidence level.

2) All-sky and cloud-free DSR trends were in the ranges 0.6 – 4.3 W m$^{-2}$/decade and -2.9 – 3.3 W m$^{-2}$/decade, respectively. Half of the trends were significant at the ≥90 % confidence level.

3) All-sky and cloud-free DLR trends were all positive and in the ranges 0.9 – 4.3 W m$^{-2}$/decade and 2.4 – 5.4 W m$^{-2}$/decade, respectively. All but one trend was significant at the ≥90 % confidence level.

4) The estimated net radiative cooling due to clouds, the CRE, decreased in magnitude by 0.9 – 3.1 W m$^{-2}$/decade over the 1996 – 2015 period, although no trends were significant at the 95 % confidence level. This decrease in CRE is probably caused by variations in macrophysical and microphysical cloud properties. However, it is not possible to determine and quantify, which cloud properties have changed and contributed to the decrease in CRE due to the lack of corresponding continuous long-term observations.

5) Between 10 and 70 % of the increase in DLR, depending on location, is explained by factors other than $T_{2m}$ and IWV. An increase in cloud cover by high level clouds appears to be consistent with these observations. However, it is not possible to quantify or verify changes in cloud properties in further detail as cloud cameras, ceilometers, lidar, etc. have only been installed to varying degrees at the four SACRaM stations in recent years.

6) Trends in AOD at each station during the 1996 – 2012 period were insignificant, and hence their impact on the observed trends of surface radiative fluxes was considered to be negligible.

Although accurate DSR and DLR time-series have been available for more than 20 years in Switzerland, the detection of trends with high confidence remains difficult due to the natural variability and measurement uncertainty in surface radiation and cloud properties. Therefore, it is crucial to continue providing facilities to maintain such radiation observations of the highest possible accuracy, which allow changes in radiation and clouds to be reliably assessed. A reduction in quality, data gaps or discontinuation of these observations may hamper the accurate detection of any trend, and thus hamper climate monitoring. Regarding the observations of clouds, it is essential to apply and develop methods, which can be used during night and day to reliably detect clouds. In addition, these methods should be capable of determining macrophysical and microphysical cloud properties, e.g., cloud type in order to verify hypotheses from observed radiation data. Such methods include lidar and cloud radar, which are limited, however, to a few super-sites due their high costs. Alternatively, visible and infrared sky cameras are promising methods, which would allow basic cloud properties to be monitored on a more widespread basis.

**Data availability.** The data sets analysed in this study are available from the corresponding author upon request (stephan.nyeki@pmodwrc.ch).

**Author contributions.** SW and JG designed the study. LV provided measurement data. SN performed the data analysis, interpreted the results and wrote the manuscript with help from SW. All other co-authors contributed by commenting and revising the paper.

**Acknowledgments.** The author was kindly financed through the Swiss GAW CRUX project by MeteoSwiss. IWV data were obtained from the STARTWAVE database, which is managed by the Institute of Applied Physics, University of Bern. The authors thank the reviewers for their constructive comments and thoughtful suggestions.

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

**Table 1: Summary of selected parameters during all-sky conditions for the 1996 – 2015 period at the four SACRaM stations (ordered by ascending altitude: LOC = 367 m, PAY = 491 m, DAV = 1594 m; JFJ = 3580 m). Average values constructed from 10-min data are shown with the standard deviations in brackets.**

| Parameter | Station | Spring (MAM) | Summer (JJA) | Autumn (SON) | Winter (DJF) | Annual |
|---|---|---|---|---|---|---|
| Temperature, $T_{2m}$ | LOC | 12.8 (5.4) | 21.3 (4.2) | 12.8 (5.3) | 4.3 (3.5) | 12.8 (7.6) |
| (°C) | PAY | 9.6 (6.0) | 18.3 (5.1) | 9.8 (5.8) | 1.2 (4.4) | 9.8 (8.1) |
| | DAV | 3.3 (6.2) | 12.0 (5.1) | 4.6 (6.2) | −4.3 (5.1) | 4.0 (8.1) |
| | JFJ | -8.6 (5.1) | -0.3 (3.7) | -5.4 (5.4) | -12.6 (5.2) | -6.7 (6.6) |
| Specific humidity | LOC | 5.6 (2.3) | 10.3 (2.5) | 7.0 (2.6) | 3.4 (1.2) | 6.6 (3.3) |
| (g kg$^{-1}$) | PAY | 5.7 (1.9) | 9.6 (1.9) | 6.8 (2.2) | 3.7 (1.1) | 6.5 (2.8) |
| | DAV | 4.1 (1.4) | 7.5 (1.6) | 5.0 (1.8) | 2.6 (1.0) | 4.8 (2.3) |
| | JFJ | 2.3 (1.2) | 4.3 (1.5) | 2.7 (1.4) | 1.5 (0.8) | 2.7 (1.6) |
| IWV | LOC | 15.8 (6.2) | 28.0 (7.3) | 19.0 (7.2) | 10.0 (4.1) | 18.1 (9.0) |
| (mm) | PAY | 14.4 (5.4) | 24.4 (6.1) | 17.3 (6.3) | 10.0 (4.2) | 16.4 (7.6) |
| | DAV | 9.5 (3.6) | 17.0 (4.1) | 11.3 (4.3) | 6.4 (2.8) | 11.0 (5.4) |
| | JFJ* | 4.6 (2.2) | 8.3 (2.8) | 5.3 (2.7) | 3.0 (1.5) | 5.3 (3.0) |
| DSR | LOC | 191.2 (275.7) | 249.0 (317.1) | 111.0 (193.1) | 72.0 (134.1) | 156.2 (250.8) |
| (W m$^{-2}$) | PAY | 184.7 (261.6) | 242.7 (304.5) | 100.5 (176.5) | 53.7 (106.5) | 145.7 (237.5) |
| | DAV | 204.0 (283.4) | 229.8 (309.6) | 119.3 (199.6) | 78.0 (142.0) | 158.4 (251.3) |
| | JFJ | 235,6 (311.4) | 265.9 (338.7) | 140.4 (224.7) | 84.9 (153.5) | 182.4 (277.5) |
| DLR | LOC | 310.5 (39.5) | 360.6 (29.7) | 320.7 (42.0) | 269.6 (37.1) | 315.6 (49.4) |
| (W m$^{-2}$) | PAY | 304.5 (38.6) | 348.0 (30.1) | 318.2 (37.4) | 285.2 (37.6) | 314.1 (42.7) |
| | DAV | 276.4 (39.9) | 319.6 (30.2) | 282.7 (40.2) | 243.5 (41.6) | 280.4 (46.9) |
| | JFJ | 224.6 (50.8) | 260.9 (45.5) | 231.1 (49.8) | 201.1 (50.3) | 229.4 (53.5) |
| Fractional cloud | LOC | 0.56 | 0.54 | 0.58 | 0.51 | 0.55 |
| cover, FCC | PAY | 0.64 | 0.60 | 0.74 | 0.81 | 0.70 |
| | DAV | 0.70 | 0.72 | 0.66 | 0.64 | 0.68 |
| | JFJ | 0.70 | 0.74 | 0.64 | 0.61 | 0.67 |

*IWV at JFJ based on parameterisation (Leckner, 1978) rather than GNSS measurements. See text for discussion.

**Table 2: Similar to Table 1 except for cloud-free conditions.**

| Parameter | Station | Spring (MAM) | Summer (JJA) | Autumn (SON) | Winter (DJF) | Annual |
|---|---|---|---|---|---|---|
| Temperature, $T_{2m}$ | LOC | 13.9 (5.7) | 22.6 (4.3) | 13.5 (5.9) | 4.9 (4.0) | 13.6 (8.1) |
| (°C) | PAY | 10.1 (7.1) | 19.5 (5.8) | 10.9 (6.6) | 0.4 (4.9) | 11.9 (8.9) |
| | DAV | 3.4 (7.1) | 13.5 (5.6) | 5.3 (6.8) | −5.1 (5.6) | 3.8 (9.1) |
| | JFJ | -7.9 (5.1) | 1.0 (3.5) | -3.9 (5.4) | -11.3 (5.0) | -6.0 (6.6) |
| Specific humidity | LOC | 4.7 (2.1) | 9.6 (2.6) | 6.2 (2.6) | 2.9 (1.0) | 5.9 (3.3) |
| (g kg$^{-1}$) | PAY | 5.3 (1.9) | 9.5 (2.0) | 6.9 (2.3) | 3.3 (0.9) | 6.8 (2.9) |
| | DAV | 3.5 (1.3) | 7.2 (1.5) | 4.6 (1.7) | 2.1 (0.8) | 4.2 (2.3) |
| | JFJ | 1.7 (1.0) | 3.4 (1.5) | 2.0 (1.2) | 1.1 (0.6) | 2.0 (1.3) |
| IWV | LOC | 12.7 (5.5) | 25.0 (7.0) | 15.6 (6.4) | 7.9 (3.1) | 15.1 (8.5) |
| (mm) | PAY | 12.0 4.8) | 22.1 (5.4) | 15.3 (5.4) | 7.7 (3.2) | 15.3 (7.2) |
| | DAV | 7.4 (3.1) | 15.3 (3.6) | 9.2 (3.6) | 4.5 (2.0) | 8.8 (4.9) |
| | JFJ* | 3.3 (1.8) | 6.5 (2.8) | 4.0 (2.2) | 2.2 (1.3) | 3.8 (2.5) |
| DSR | LOC | 270.2 (320.1) | 358.9 (356.3) | 177.1 (238.4) | 99.6 (159.8) | 229.3 (295.7) |
| (W m$^{-2}$) | PAY | 257.7 (313.4) | 328.3 (347.0) | 166.2 (237.3) | 99.7 (160.5) | 206.8 (304.3) |
| | DAV | 265.9 (336.2) | 309.4 (362.2) | 176.1 (246.0) | 109.4 (176.5) | 216.2 (294.2) |
| | JFJ | 299.5 (388.7) | 354.7 (408.4) | 190.3 (270.2) | 116.6 (187.0) | 244.9 (328.1) |
| DLR | LOC | 283.1 (31.3) | 344.1 (26.7) | 290.1 (33.3) | 241.9 (17.9) | 289.3 (46.3) |
| (W m$^{-2}$) | PAY | 274.8 (31.3) | 329.9 (26.6) | 286.7 (32.1) | 234.0 (19.9) | 289.6 (43.5) |
| | DAV | 237.5 (28.2) | 291.5 (22.2) | 249.4 (28.2) | 204.5 (20.6) | 243.4 (39.8) |
| | JFJ | 167.6 (22.3) | 207.9 (18.0) | 182.3 (23.6) | 151.9 (19.4) | 175.2 (29.1) |

*IWV at JFJ based on parameterisation (Leckner, 1978) rather than GNSS measurements. See text for discussion.

**Table 3: Trend analyses (linear least squares, LLS, and Sen's methods) of selected parameters for the 1996 – 2015 period during all-sky and cloud-free conditions at all four stations. Trend values in italic (bold) are significant at the 90 % (95 %) level. The 90 % confidence interval of each trend is shown in square brackets for the LLS method.**

| Parameter | Station | All-sky LLS method unit/decade | All-sky Sen's slope unit/decade | Cloud-free LLS method unit/decade | Cloud-free Sen's slope unit/decade |
|---|---|---|---|---|---|
| Temperature, $T_{2m}$ (°C) | LOC | **0.43** [**±0.25**] | **0.53** | **0.54** [**±0.27**] | **0.66** |
| | PAY | *0.35* [*±0.29*] | **0.50** | **0.59** [**±0.33**] | **0.79** |
| | DAV | 0.30 [±0.32] | 0.44 | *0.48* [*±0.38*] | **0.61** |
| | JFJ | 0.34 [±0.32] | 0.43 | 0.20 [±0.38] | 0.16 |
| Specific humidity ($g\ kg^{-1}$) | LOC | **0.19** [**±0.13**] | **0.18** | 0.14 [±0.15] | 0.12 |
| | PAY | **0.18** [**±0.11**] | **0.19** | **0.23** [**±0.14**] | **0.18** |
| | DAV | *0.08* [*±0.07*] | 0.08 | *0.10* [*±0.09*] | 0.10 |
| | JFJ | **0.14** [**±0.06**] | **0.14** | **0.19** [**±0.07**] | **0.19** |
| IWV (mm) | LOC | 0.37 [±0.56] | 0.42 | 0.36 [±0.60] | 0.31 |
| | PAY | 0.41 [±0.48] | **0.80** | *0.58* [*±0.55*] | **1.03** |
| | DAV | **0.63** [**±0.31**] | **0.89** | **0.79** [**±0.37**] | **1.18** |
| | JFJ* | **0.24** [**±0.11**] | **0.26** | **0.26** [**±0.13**] | **0.25** |
| DSR ($W\ m^{-2}$) | LOC | *4.3* [*±3.3*] | **5.5** | 3.3 [±3.8] | **3.8** |
| | PAY | *3.4* [*±3.2*] | *3.4* | **10.6** [**±4.0**]** | **10.0**** |
| | DAV | 0.6 [±2.9] | 0.2 | 3.1 [±5.1] | 3.5 |
| | JFJ | *3.6* [*±2.7*] | 2.2 | **-9.5** [**±4.8**]*** | **-10.3**** |
| DLR ($W\ m^{-2}$) | LOC | *2.5* [*±1.9*] | **2.5** | **2.9** [**±1.8**] | **3.2** |
| | PAY | 0.9 [±1.6] | 0.9 | **2.4** [**±1.9**] | *2.5* |
| | DAV | **2.7** [**±1.5**] | **3.2** | **4.8** [**±1.7**] | **5.8** |
| | JFJ | **4.3** [**±2.1**] | **5.9** | **5.4** [**±1.6**] | **5.9** |

*IWV at JFJ based on parameterisation (Leckner, 1978) rather than GNSS measurements. See text for discussion. **Trends for 1996 – Dec. 2011 are 2.9 and 3.0 W m$^{-2}$/decade (none significant) for the LLS and Sen's methods, respectively. ***Trends for 1996 – Dec. 2007 are -2.9 and -1.7 W m$^{-2}$/decade (none significant), respectively.

**Table 4: Long-term average values of the shortwave and longwave cloud effects (SCE and LCE, respectively), and cloud radiative effect (CRE) for the 1996 – 2015 period at all four stations. The one sigma uncertainties are shown in brackets.**

| Station | SCE (W m$^{-2}$) | LCE (W m$^{-2}$) | CRE (W m$^{-2}$) |
|---------|------------------|------------------|------------------|
| LOC | -47.7 (±6.1) | 23.3 (±2.8) | -24.4 (±5.1) |
| PAY | -71.9 (±5.5) | 31.1 (±2.5) | -40.8 (±4.4) |
| DAV | -72.8 (±5.6) | 32.1 (±3.0) | -40.7 (±3.9) |
| JFJ | -54.2 (±3.9) | 49.9 (±5.6) | -4.3 (±5.3) |
| Average | -61.6 (±7.4) | 34.1 (±3.7) | -27.6 (±5.4) |

**Table 5: Trend analysis of the shortwave and longwave cloud effects (SCE and LCE), and cloud radiative effect (CRE) for the 1996 – 2015 period at all four stations. Trends in italic (bold) are significant at the 90 % (95 %) confidence level. The 90 % confidence interval of each trend is shown in square brackets.**

| Station | SCE (W m$^{-2}$/decade) | | LCE (W m$^{-2}$/decade) | | CRE (W m$^{-2}$/decade) | |
|---------|------|-------------|------|-------------|------|-------------|
| | LLS | Sen's Slope | LLS | Sen's Slope | LLS | Sen's Slope |
| LOC | 3.6 [±3.6] | 2.2 | -0.7 [±1.4] | -0.5 | 2.9 [±3.1] | 2.3 |
| PAY | *3.8 [±3.6]* | *3.8* | -0.6 [±1.4] | -0.9 | *3.1 [±2.7]* | *2.3* |
| DAV | -0.1 [±3.6] | -0.4 | 1.0 [±1.5] | 1.4 | 0.9 [±2.6] | 1.3 |
| JFJ | 0.1 [±3.1] | -0.5 | 2.4 [±2.2] | 2.8 | 2.5 [±2.6] | 2.4 |

**Table 6: Trend analysis of the longwave discrepancies during cloud-free conditions for the 1996 – 2015 period at all four stations. Trend values in italic (bold) are significant at the 90 % (95 %) level. Percentage values in brackets correspond to the contribution of the discrepancy to the overall trends in Table 3.**

| Station | Longwave | |
| | LLS method (W m$^{-2}$/decade) | Sen's slope (W m$^{-2}$/decade) |
| --- | --- | --- |
| LOC | 0.3 (10 %) | 0.3 (9 %) |
| PAY | 1.2 (51 %) | 1.3 (46 %) |
| DAV | **3.4 (70 %)** | **3.1 (60 %)** |
| JFJ | **3.8 (70 %)** | **4.6 (78 %)** |

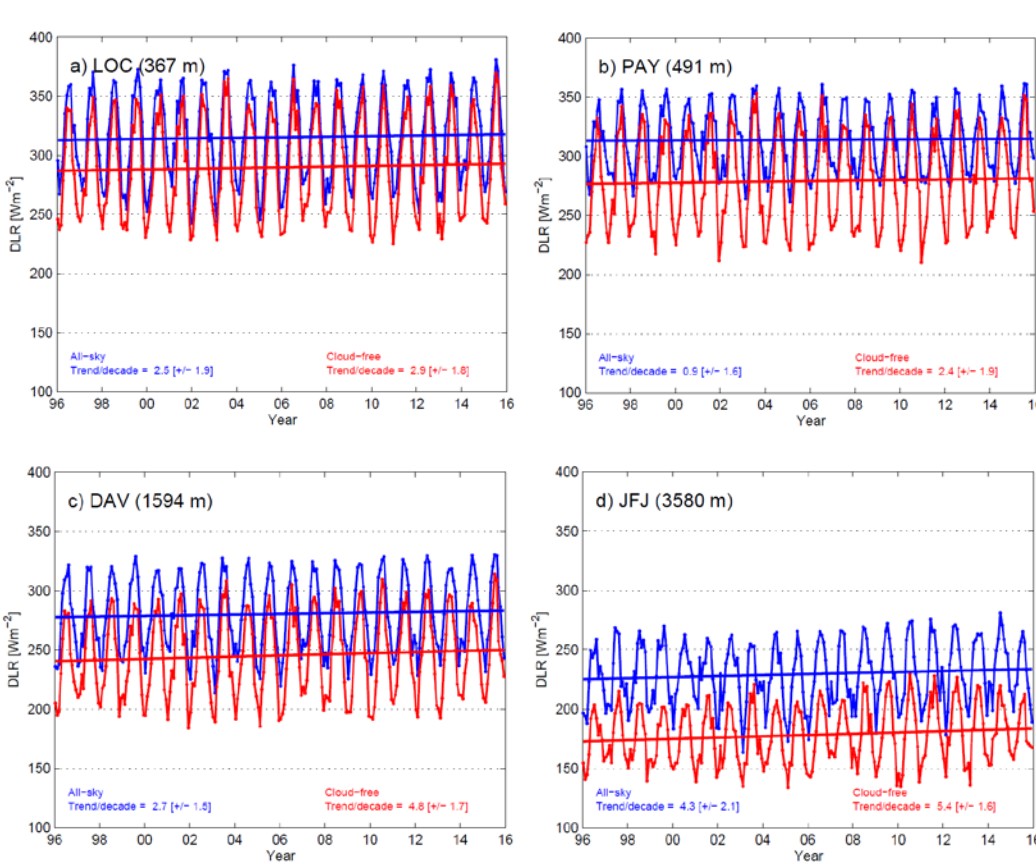

**Figure 1: Monthly average DLR values during all-sky (blue) and cloud-free (red) conditions at: a) Locarno, b) Payerne, c) Davos, and d) Jungfraujoch. Each panel shows trend results from linear least squares analysis in Table 3. Values in square brackets represent the 90 % confidence interval of the trend. Scales are similar to aid the comparison.**







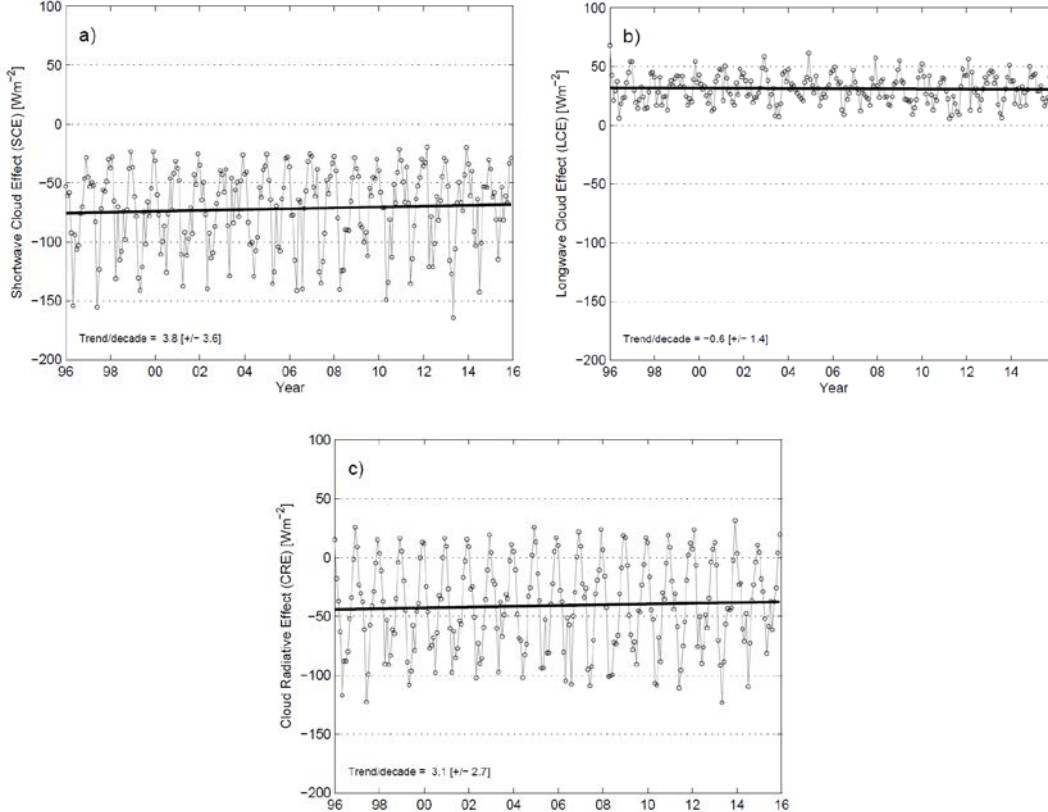

**Figure 2: Time series of monthly average: a) SCE, b) LCE and c) CRE values at Payerne (PAY). Values in square brackets represent the 90 % confidence interval of the trend.**







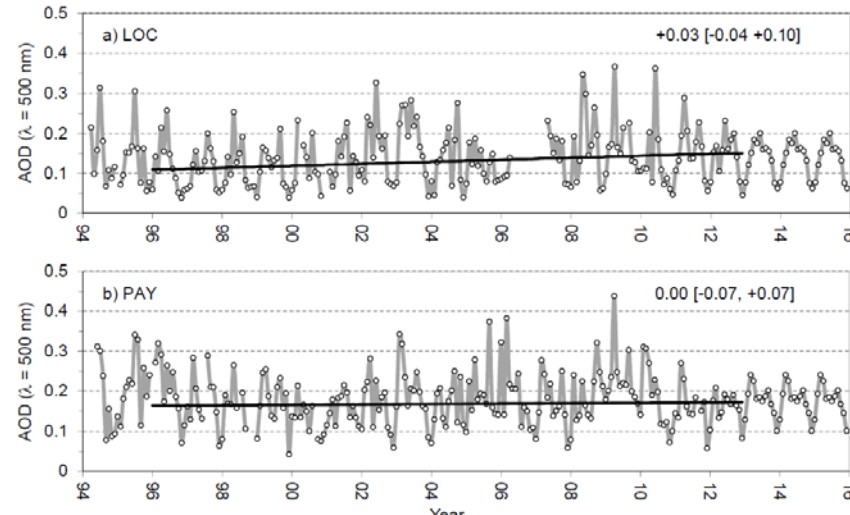

**Figure 3: Time series of monthly AOD averages ($\lambda$ = 500 nm) available since 1994 at: a) LOC and b) PAY. The 1996 – 2012 decadal trend is shown in the top right-hand corner where values in brackets represent the upper and lower bounds of the 90 % confidence**
10    **interval. Trend values are only calculated for observations during 1996 – 2013, and a climatology (1996 – 2012) was used for the 2013 – 2015 period. See text for further details.**

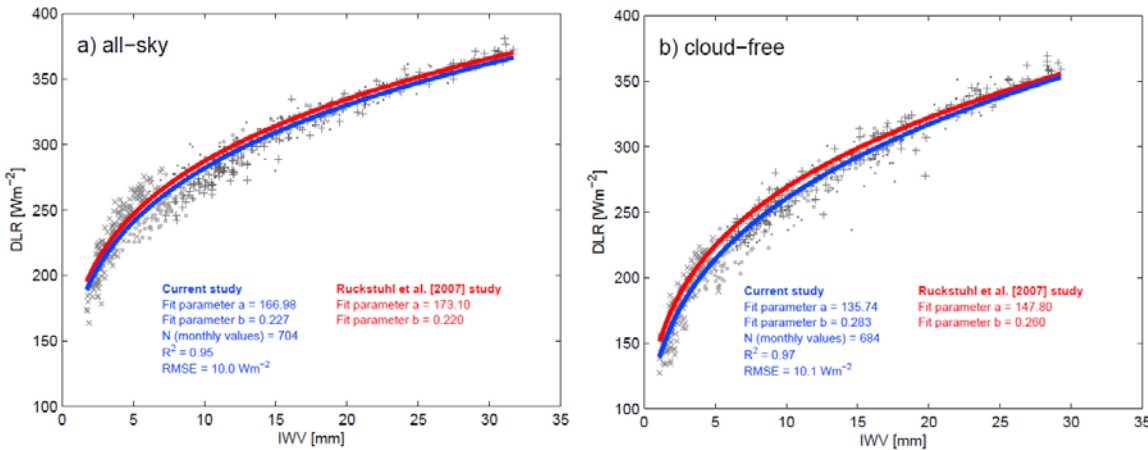

**Figure 4: Monthly average DLR at all four stations (symbols: LOC = plus symbols, PAY = closed circles, DAV = open circles, and JFJ = crosses) for the 2000 – 2015 period versus IWV values during: a) all-sky conditions, and b) cloud-free conditions.**