# Peer review of "Trends in surface radiation and cloud radiative effect at four Swiss sites for the 1996 - 2015 period"

_Atmospheric Chemistry and Physics, 2018_

## Referee Comment (RC1) · Anonymous Referee #1 · 17 May 2019

Please see review uploaded as a supplementary file.

Please also note the supplement to this comment:
https://www.atmos-chem-phys-discuss.net/acp-2018-1096/acp-2018-1096-RC1-supplement.zip

---

## Referee Comment (RC2) · Anonymous Referee #2 · 18 May 2019

This study used ground-based observations of shortwave and longwave fluxes associated with meteorological parameters under all-sky and cloud-free conditions, spanning over the period from 1996 to 2015 at four stations in Switzerland, to analyze the trends of downward shortwave and longwave radiation at the surface and consequent the trend of the cloud radiative effect.

It is important to investigate the trends of radiation and cloud radiative effect at the surface and put forward the possible explanations that account for the phenomenon. However, there are some weakness that requires more supporting material. The clouds identification methods are not accurate which may induce a contamination of radiation

fluxes, since a slight change of cloud cover may significantly influence the radiation fluxes at the surface. Aerosol burden also has significant effects on shortwave radiation at the surface. Using climatological average of AOD to fill in the missing data of AOD during 2013 through 2015 may cause an artificial error in the trend analysis. Without any detailed description of how clouds change, it is quite arbitrary and blurry to infer the relationship between the variations of CRE and clouds.

Specific comments: 1. Page 5 and line 25: 'In contrast, the specific humidity and IWV are higher during all-sky conditions which in turn results in higher DLR values.', please mention the source of the specific humidity data? 2. Page 5 and line 30 : 'IWV at JFJ was based on a widely-used parameterization using T 2m and relative humidity', where is relative humidity data from and what is the accuracy of this parameterization? Please add description of these in the section of methods and data. 3. Page 5 and line 35: 'which are probably associated with synoptic scale weather patterns', what kind of weather patterns are they? 4. Page 6 and line 25: 'the DSR trend at PAY is not monotonic but steeply', as DSR has obvious changes, could you show its trend at four sites like Figure 1? 5. Page 7 and line 20: what are the '$\sim<$' and '$-<$'? 6. Page 8 and line 15: 'suggesting that a decrease in fractional cloud cover or a different cloud type has occurred during the 1996 – 2015 period', there are many factors may affect cloud radiative effects such as cloud height and optical depth. A decrease of CRE magnitude doesn't mean there can be the decrease of cloud fraction. What are the variations of different clouds during the 1996 – 2015 period?

---

## Author Comment (AC1) · 20 Jul 2019

20 July 2019

Dear Editor and Referees.

Please find our corrections below. We thank both Referees for their thoughtful comments and detailed corrections. It has taken longer than anticipated to correct our paper, but the effort has definitely been worthwhile. We therefore hope that we have answered the questions as best as possible and that the latest version meets with the Referees approval.

[Figure]

During the revision, we came across an interesting review paper by Pepin et al. (2015), published in Nature Climate Change. Two small paragraphs have been added to the new paper on page 7 line 21 and page 9 line 41.

Pepin, N., Bradley, R. S., Diaz, H. F., Baraër, M., Caceres, E. B., Forsythe, N., Fowler, H., Greenwood, G., Hashmi, M. Z., Liu, X. D. and Miller, J. R.: Elevation-dependent warming in mountain regions of the world, Nature Climate Change, 5, 424, doi: 10.1038/nclimate2563, 2015.

Best Regards Stephan Nyeki

PMOD/WRC Davos, Switzerland

âĞČ Referee 1: Comments

Major comments:

The attention the authors give to the details and quality of the radiometric measurements is unusual and much appreciated. The use of multiple techniques to evaluate continuity of the measurement time series and the significance of results is also laudable.

Answer: Thank you.

Minor comments:

Referee comment: p. 1, line 12. Technically, you can't measure "radiation," only some property of radiation, such as intensity or wavelength. I believe that in this case irradiance (or "flux") is meant.

Answer: Rather than change the whole manuscript, and in keeping with terminology in our field, we would prefer to use the term "radiation" as well as flux when referring to shortwave and longwave radiation/flux.
* * *
[Figure]

Referee comment: p. 1, line 16. Insert "surface".

Answer: Inserted
* * *
Referee comment: p. 1, line 33. Over what period?

Answer: The sentence has been updated to: "DLR has also been observed to increase during the 1973 – 2008 period (Wang and Liang, 2009) and since the 1990s (Wild, 2016b), although . . .".
* * *
Referee comment: p. 1, line 30. You just said that the record began in the late 1980s...?

Answer: The sentence has been updated to: "DSR, from earlier less reliable measurements, over Europe . . .".
* * *
Referee comment: p. 1, line 39. Insert "longwave".

Answer: Inserted.
* * *
Referee comment: p. 1, line 39. Change "which" to "that".

Answer: As a native British English speaker, I would argue that "which" is ok. As our ACP paper will undoubtedly be language-corrected before publication, may we recommend that ACP decide?
* * *
Referee comment: p. 2, line 3. I think you mean that the CRE decreased, not the trend.

Answer: the text has been changed and the citation year corrected.

"CRE decreased at the same four Swiss stations by up to 7.5 W m-2 for the 1996 – 2010 period (Wacker et al., 2013), . . . ".
* * *
Referee comment: p. 2, line 9. Include the word "total".

Answer: Inserted.
* * *
Referee comment: p. 2, line 13. Please add a citation where the guidelines can be found.

Answer: The original sentence was: "Measurements were conducted according to BSRN guidelines".

This has been slightly changed and corrected to:

"Measurements were conducted according to BSRN guidelines, detailed in a later report by McArthur (2005)."

McArthur, B.: Baseline Surface Radiation Network (BSRN), Operations Manual Version 2.1, WCRP 121, WMO/TD-No. 1274, 2005.

Referee comment: p. 2, line 14. Are or were, not "included." This implies that the stations listed are a subset of the four that continued operation.

Answer: The sentence has been changed to: "The remaining stations are . . .".
* * *
Referee comment: p. 2, line 21. Several comments. We have revised the paragraph to include instrumental details.

New i) Pyrgeometers in the ASRB network were all unshaded, and hence a correction for solar heating of the instrument was applied using the method described by Dürr

(2004). In contrast, it was unnecessary to correct DLR data from the SACRaM network. These were either shaded pyrgeometers (Precision Infrared type, PIR, Eppley Inc., USA) or unshaded pyrgeometers CG(R)4, Kipp & Zonen, Netherlands). As CG(R)4 pyrgeometers are less affected by heating effects or by longwave irradiance in the direct beam of the sun (Meloni et al., 2012; Gröbner et al., 2018), no correction is necessary.
* * *
Referee comment: p. 2, paragraph starting "iv": This paragraph is unclear. Does it mean that the measurements from PMOD are wrong and that those from BSRN are right? Which data requires correction?

Answer: Short and longwave reference scales may need to be revised in the future, depending on the decision of various scientific bodies. BSRN short and longwave time series are referenced to these scales, and hence may also need to be revised. Using the terms "right" or "wrong" would not be appropriate here.

We feel that our original text reflects the current situation quite well but have made subtle changes to the paragraph, which is now:

"iv) The PMOD/WRC hosts the World Standard Group (WSG) of pyrheliometers and the WISG, as mentioned above. These provide the reference scales for shortwave and longwave radiation measurements, respectively. However, as a note of interest, several studies have determined that their reference scales may need to be revised in the future (Fehlmann et al., 2012; Gröbner et al., 2014). The WSG scale currently overestimates by +0.3 %, and a linear correction could be applied in a straightforward manner. However, the WISG scale underestimates longwave fluxes, which will require a non-linear correction depending on a number of factors (e.g., raw signal data, etc.), as reported by Gröbner et al. (2014) and Nyeki et al. (2017). The latter study determined that corrections were in the ranges 1 to 4 W m-2 for all-sky DLR and 5 to 7 W m-2 for cloud-free DLR when based on available data from three BSRN stations and Davos

(i.e., PMOD/WRC), which have the longest time series. Such corrections are beyond the scope of the present study, and are currently being debated within the community. A possible future correction of the SACRaM DSR time series should have no effect on the trend analyses in this study while corrected DLR time series could marginally affect the trends depending on the degree of cloudiness at each station."
* * *
Referee comment: p. 3, line 1, insert "as".

Answer: Inserted
* * *
Referee comment: p. 3, paragraph starting "v": Are these uncertainties for instantaneous measurements?

Answer: The uncertainties for pyranometers that are cited in Vuilleumier et al. (2014) are for high intensity (1000 W m-2) 1-min averages as defined by BSRN, while for low intensity (50 W m-2), the uncertainties are lower, although not proportionally. The sentence has been changed to:

"The uncertainty of pyranometer measurements is estimated to be in the range 18 – 23 W m-2 for 1-min average values (Vuilleumier et al., 2014)."
* * *
Referee comment: p. 3, line 1, Why +/- here but not for pyranometers? This is confusing.

Answer: Actually the range is not +/- 4 Wm-2 for all-sky DLR and +/- 7 Wm-2 for cloud-free DLR but between +1 and +4 Wm-2 and +5 and +7 Wm-2 for all-sky and cloud-free DLR, respectively. It may also be that you are referring to the pyranometer range in "%" and the pyrgeometer range in "Wm-2". What can we say except that this is how the community reports these ranges.

However, for clarity, the original sentence has been changed from:

"The latter study determined that corrections lay in the ranges $\sim 1 - 4$ W m-2 for all-sky DLR and $\sim 5 - 7$ W m-2 for cloud-free DLR when based on available data from four BSRN stations having the longest time series".

to:

"The latter study determined that corrections were in the ranges +1 to 4 W m-2 for all-sky DLR and +5 to 7 W m-2 for cloud-free DLR when based on available data from three BSRN stations and Davos (i.e., PMOD/WRC), which have the longest time series."
* * *
Referee comment: p. 3, line 12: What meteorological data?

Answer: "T2m, RH, and pressure" inserted.
* * *
Referee comment: p. 3, line 17: Move "was considered".

Answer: As a native British English speaker, I would argue that our version is correct. As our ACP paper will undoubtedly be language-corrected before publication, may we recommend that ACP decide?
* * *
Referee comment: p. 3, line 19: For both the all-sky and clear-sky data?

Answer: We would argue that this is implicit, and the sentence therefore does not have to be lengthened with more detail.
* * *
Referee comment: p. 3, line 20: Was a sampling threshold also applied to the clear-sky data?

Answer: The original sentence was: "A monthly average was accepted for all-sky conditions if ≥75% of data were available for each month"

This has been changed to:

"A monthly average was accepted for all-sky conditions if ≥75 % of data were available for each month, while no sampling threshold was applied to cloud-free data due to the smaller dataset after application of the cloud filter."
* * *
Referee comment: p. 3, line 22: Add "determination of".

Answer: Added.
* * *
Referee comment: p. 3, line 36: Change "that" to "these" or "their".

Answer: Sorry, but as a native British English speaker, I would argue that our version is correct. As our ACP paper will undoubtedly be language-corrected before publication, may we recommend that ACP decide.
* * *
Referee comment: p. 4, line 3: What is the source of these reference values?

Answer: The original sentence has been changed from:

"After the pre-processing of images (Aebi et al., 2017), a color ratio (the sum of the blue to green ratio plus the blue to red ratio) is calculated per pixel (Wacker et al., 2015) and compared to a reference value (2.2 in Davos and 2.5 in Payerne)."

to:

"After the pre-processing of images (Aebi et al., 2017), a color ratio (the sum of the blue to green ratio plus the blue to red ratio) is calculated per pixel (Wacker et al.,

2015) and compared to empirically determined reference values (2.2 in Davos and 2.5 in Payerne), which are based on a large database of sky camera images."
* * *
Referee comment: p. 4, line 5: Change "attributed to" to "categorised as".

Answer: We consider "attributed to" to be ok, but have changed this to "categorised as".
* * *
Referee comment: p. 4, line 21: Monthly or weekly climatological values? Referee comment: p. 4, lines 20-21: Could you at least give the general basis for the derivation of AOD values - sun photometer measurements? satellite measurements?

Answer: We would prefer not to add too much extra detail as two references have been cited. However, the sentence has been changed to the following:

"AOD from sun-photometers at each of the four sites was derived using procedures and data published previously (Nyeki et al., 2012; Kazadzis et al., 2018). AOD data (1 min.) was only available for Jan. 1994 – Dec. 2012, which was used to construct an AOD climatology for the Jan. 2013 – Dec. 2015 missing period. While this may introduce an error in the AOD trend, a large change is not expected as the measured time series is 18 years long. "
* * *
Referee comment: p. 4, line 30: What is the definition of the "effective atmospheric boundary layer temperature" and how is the value obtained?

Answer: Definition and retrieval of the effective atmospheric boundary layer temperature are described in detail by Gröbner et al. (2009). The following sentence has been added:

"TABL represents the effective radiating temperature of water vapour in the atmospheric boundary layer, and is derived by using two co-located pyrgeometers: one standard pyrgeometer sensitive to the $3-50$ $\mu$m wavelength range and another modified one, which is sensitive in the $8-14$ $\mu$m range."
* * *
Referee comment: p. 4, line 33: It would be helpful to have the final form of the equation (as used) written out here?

Answer: As T2m = TABL, the full equation would be the same as the existing Eq. 3. We would therefore prefer not to add another identical equation. However, we have more clearly stated that T2m = TABL in the following sentence:

"Setting T2m = TABL, as well as use of the Prata parameterisation was considered by . . ."
* * *
Referee comment: p. 5, line 3: Are the values of a and b the same for all times and locations?

Answer: The preceding sentence to line 3 was:

"A power-law of the following form was found for this DLR-IWV parameterisation when data from all four stations was combined:"

This has been changed to the following in order to answer the referee's comment:

"A power-law of the following form was found for this DLR-IWV parameterization when data from all four stations and time periods was combined into a single equation:"
* * *
Referee comment: p. 5, line 8: I thought Eqn. 4 WAS the method?

Answer: Eq. 4 is an alternative parameterisation used for DLRsim cloud-free. We state

this on p.4 line 36 of the original manuscript, and therefore feel that no changes are necessary.
* * *
Referee comment: p. 5, line 12: 1998 is before 2011, so how is this a "forerunner"?

Answer: "Forerunner" refers to the forerunner of the present study. To avoid confusion, we have removed the word.
* * *
Referee comment: p. 5, line 14: Change to "that"

Answer: As a native British English speaker, I would argue that our version is correct. As our ACP paper will undoubtedly be language-corrected before publication, we would recommend that ACP decide.
* * *
Referee comment: p. 5, line 15: It sounds like you mean that the trend is increasing or decreasing over time. Do you really mean a positive or negative trend?

Answer: The text has been changed to "positive or negative trend".
* * *
Referee comment: p. 5, line 16: Shouldn't these procedures have been applied before the trend analysis?

Answer: This was, of course, the case. However, as it may not be clear from the text that this occurred, we have changed the sentence from:

"In order to check the homogeneity of the time series, three statistical tests were applied:"

to

"Before these trend tests were applied, the homogeneity of the time series was checked using three tests".
* * *
Referee comment: p. 5, line 27: They're similar but they're different (following lines)?

Answer: Thank you. The text had not been updated from a previous version. The sentence has now been changed from:

"All-sky values in Table 1, not previously reported, are similar to cloud-free values in Table 2 for the 1996 – 2007 period reported by Wacker et al. (2011).

to:

"All-sky values in Table 1 have not been previously reported while cloud-free values in Table 2 are similar to values reported by Wacker et al. (2011) for the 1996 – 2007 period."
* * *
Referee comment: p. 5, lines 33-34: Isn't there snow or ice at the other locations?

Answer: The original sentence was:

"ii) the IWV retrieval algorithm is unable to adequately correct for the influence of snow and ice on the GNSS antenna signal".

The sentence now includes the word "persistent", reflecting the conditions at JFJ.

"ii) the IWV retrieval algorithm is unable to adequately correct for the persistent influence of snow and ice on the GNSS antenna signal."
* * *
Referee comment: p. 5, lines 35-40: I'm not convinced that means of 0.7, 0.68, and 0.67 are significantly different. However, the different seasonal pattern at PAY looks

interesting. Can you comment on that?

Answer: We are not sure what is meant by the Referee's question. The original text does not discuss the above-mentioned average values in the way the Referee states. In particular, we do not state that means of 0.7, 0.68, and 0.67 are significantly different. We indicate that FCC is reduced South of the Alps, and Payerne shows the highest FCC as a result of more persistent stratus cloud cover.

We have changed the sentence as follows:

"The clearest conditions occur at Locarno (lee location, south of the Alps) with an average FCC value = 0.55 while the cloudiest conditions occur at PAY (plateau location, north of the Alps) with FCC = 0.70 as a result of more persistent stratus cloud cover particularly during winter time when low cloud type stratus nebulosus regularly covers the Swiss Plateau."
* * *
Referee comment: p. 6, lines 10-11: Why should the frequency of occurrence of clear-sky conditions affect the average clear-sky DSR value?

Answer: Thank you for highlighting this. The sentence has now been changed to:

"Again, the climatology at each station also has an influence as the cloud-free annual average DSR at LOC (229.3 W m-2) is higher than at PAY (206.8 W m-2) and DAV (216.2 W m-2)."
* * *
Referee comment: p. 6, line 17: Add the word "nevertheless"

Answer: Inserted.
* * *
Referee comment: p. 6, line 17: "prior" might be better. "forerunner" makes it sounds

like these studies were long ago.

Answer: We feel that "forerunner" is an appropriate term but have changed it to "prior".
* * *
Referee comment: p. 6, line 18 Referee comment: "Also"? This is the first discussion of actual values. I would just start a new paragraph.

Answer: "Also" removed. New paragraph started.
* * *
Referee comment: p. 6, line 23: It looks to me like just as many DSR trends are significant at the 90% level as T2m trends (although fewer are significant at the 95% level).

Answer: The sentence was not formulated well. It has now been changed to:

"Trends in all-sky DSR are in the 0.6 – 4.3 W m-2/decade range with a significance at the 90% confidence level except for DAV".
* * *
Referee comment: p. 6, line 26: If it's flat, it's not an increase. Do you mean "slow"?

Answer: We feel that the term "flat increase" is an appropriate term but have changed it to "slow".
* * *
Referee comment: p. 6, line 38 and line 42: Don't you mean discontinuities? You already said there were changes/trends.

Answer: "Discontinuities" has been used instead.
* * *
Referee comment: p. 6, line 41: How do these three values relate to the four stations?

Answer: Rather than repeat numerous p-values, the values from all stations have been grouped together, and are discussed as a group with respect to each statistical test. Our original text states: "... from all four stations ...". We feel that this is clear and concise, and it is therefore unnecessary to add more detail.
* * *
Referee comment: p. 6, line 41: How can p values of 0.92 and 0.17 both indicate significance?

Answer: The term "significance" is perhaps incorrectly used here, so we have therefore omitted it. Only values p < 0.05 are significant which was not observed.

The sentenced has been moved and changed to:

"Results from the SNHT, Buishand and Pettitt homogeneity tests indicate that no time series at any station had p < 0.05, suggesting that all meteorological time series were homogeneous with no significant discontinuities due to climatic or non-climatic effects such as a change of instrument or data acquisition system, relocation, etc."
* * *
Referee comment: p. 6, lines 38-42: Please give specific results of the homogeneity tests for DSR and DLR at Davos, since text on page 2 describes differences in the instrumentation used for different time periods.

Answer: The sentence has been changed to:

"Homogeneity analyses of all meteorological parameters were then conducted to test for any discontinuities in the time series. This is only meaningful when using the full dataset i.e., for all-sky conditions as opposed to cloud-free conditions, which are a sub-set of the former. Results from the SNHT, Buishand and Pettitt homogeneity tests indicate that no time series at any station had p < 0.05, suggesting that all meteorolog-

ical time series were homogeneous with no significant discontinuities due to climatic or non-climatic effects such as a change of instrument or data acquisition system, relocation, etc."
* * *
Referee comment: p.7, line 6: Change sentence.

Answer: Changed.
* * *
Referee comment: p. 7, first full paragraph: Is it meaningful to compare trends from all these different time periods? It also seems that trends from global means and individual stations are being compared. I wouldn't expect trends from climate models to be particularly accurate. (Why would you look at two RCPs when we know what the CO2 concentrations were over the time periods of interest?) You might think about presenting results from satellite studies instead/in addition.

Answer: Unfortunately, we disagree with this statement. We are comparing observations in central Europe with BSRN observations from around the World. Hence, we consider that widening the study to include DSR and DLR from satellites would be beyond the scope of the present study in order to add more detail to a single paragraph.

We have therefore changed the original sentence from:

"Values of -61.6, 34.1 and -27.6 W m-2, respectively, are broadly similar to recently updated global average values of -56, 28 and -28 W m-2 reported by Wild et al. (2017) using BSRN data."

To:

"Interestingly, these values are similar to recently updated global average values of -56, 28 and -28 W m-2 reported by Wild et al. (2017) using BSRN observational data. The similarity is reflected by the fact that these globally averaged values are predominantly weighted by European as well as global mid-latitude sites with similar cloud climatologies."
* * *
Referee comment: p.7, line 19: The word "anomaly" isn't really appropriate here. For example, one definition is "something unusual, unexpected, or different from what normally happens" (Macmillan online dictionary). In meteorology, the term is often applied to mean differences from the long term mean, so may be confusing to readers. I would suggest "differences," "discrepancies," or "deviations," although some people might call them "errors". It might be clearer if you just said that the cloud-free estimates were validated by comparison to clear sky measurements.

Answer: The term "anomaly" has been used in other studies, but we agree that "discrepancy" is more appropriate, and have therefore changed the term throughout the manuscript.
* * *
Referee comment: p.7, line 23: This figure is never discussed?

Answer: We do actually discuss this briefly on line 23 of the original manuscript, and in the paragraphs thereafter. We have therefore added the following text to the end of the sentence:

". . . which are described further below".
* * *
Referee comment: p.7, line 26: But at JFJ, it's 0.67.....

Answer: Rather than going into more detail, we would prefer to add the word "partly" to the sentence:

"This can be partly be explained by a higher cloud frequency at these sites with FCC =

0.68 and 0.70, . . . ".
* * *
Referee comment: p.7, line 29: What are "low, negative values"? I might just cut the text from "with the latter" on.

Answer: Text removed as suggested.
* * *
Referee comment: p.7, line 30: No, SCE is defined only in terms of DSR. It's the DSR that is determined by those other factors.

Answer: Thank you for pointing this out. It is of course a mistake, which we failed to notice. The paragraph has been re-written, and is now:

"Positive trends of 3.6 and 3.8 W m-2/decade (see Table 5) are observed at LOC and PAY, respectively, which represent a decrease in the magnitude of the SCE. In contrast, SCE trends at DAV and JFJ are close to zero for both the LLS and Sen's slope methods. Neither LOC nor PAY trends are significant at the 95 % confidence level but their positive values arise from the fact that trends in DSRall-sky > DSRsim cloud-free. Apart from DSR, DSRsim cloud-free is also calculated using IWV and AOD. IWV Trends at LOC and PAY in Table 3 are slightly positive but not significant while trends in AOD for the 1996 – 2015 period are shown in Figure 3. Trends are essentially negligible at 0.03 and 0.00/decade, for LOC and PAY, respectively, while those at DAV and JFJ are similar, as shown in a previous study (Nyeki et al., 2012) and in unpublished data. Positive SCE trends at LOC and PAY are therefore mainly due to positive trends in DSRall-sky."
* * *
Referee comment: p.7, line 30: Do we expect long-term trends in the solar zenith angle?

Answer: The sentence has been changed according to the above answer.

Referee comment: p.7, line 32: Again, SCE is defined by the DSR.

Answer: This has been changed. Please see our comment further above.
* * *
Referee comment: p. 7, line 35: Why is altitude important to LCE?

Answer: Because of the water vapour content (IWV) and the spectral properties of the atmosphere / water vapour continuum in the IR: in areas with high IWV (i.e., at low altitudes in the study area), the IR spectrum is close(r) to saturation even in cloud-free conditions (except for some spectral windows of the water vapour continuum, mainly the 8-14 $\mu$m wavelength range but also some other narrower spectral bands) compared to areas with low water vapour where the IR spectrum is far from saturation in a cloud-free atmosphere (the "windows" of the water vapour continuum are "open"). Therefore, the difference between observed DLR during all-sky conditions (when there are clouds) and calculated DLR for the corresponding cloud-free conditions (i.e., the LCE) is larger in areas with lower IWV compared to areas with higher IWV (see for instance also Fig. 3 in Wacker et al.2011). Or in other words: Clouds have a smaller impact in the IR at high IWV because the water vapour masks the radiative effect of clouds. The mountain site JUN is frequently in clouds causing a saturated water vapour continuum in the IR and thus a large difference, i.e. LCE, with respect to the corresponding calculated cloud-free fluxes.

S. Wacker, J. Gröbner, D. Nowak, L. Vuilleumier, and N. Kämpfer. Cloud effect of persistent stratus nebulosus at the Payerne BSRN site. Atmospheric Research, 102, 1–9, 2011.

The original sentence has been changed from:

"Regarding the LCE, annual average values are all positive with the highest occurring at JFJ (49.9 W m-2) and the lowest at LOC (23.3 W m-2), which is partly due to their

altitudes at 3580 m and 367 m, respectively".

to:

"Regarding the LCE, annual average values are all positive with the highest occurring at JFJ (49.9 W m-2) and the lowest at LOC (23.3 W m-2). LCE decreases with decreasing altitude due to the higher water vapour content and thus higher cloud-free longwave fluxes (e.g., Wacker et al., 2011b; Aebi et al., 2017)."
* * *
Referee comment: p. 7, line 36: Change to "is".

Answer: As a native English-speaker, I would argue that our version is correct. The sentence effectively reads: "...Trends ... are ... consistent...". However, the ACP copy-editors can change this if they so wish.
* * *
Referee comment: p. 7, line 38: See prior comment about DSR and SCE.

Answer: Thank you for pointing this out. It is of course a mistake, which we failed to notice. The original sentence has been replaced with the following sentence:

"The LCE depends on a range of microphysical and macrophysical cloud properties, as mentioned in Section 1."
* * *
Referee comment: p. 8, line 1: Maybe mention that SCE dominates total CRE before this statement? Referee comment: p. 8, line 2: How does the reduction in daylight hours during the winter affect CRE at DAV and PAY more than at LOC and JFJ? Aren't all the sites at about the same latitude? Also, does this sentence only pertain to winter?

Answer: The sentence is somewhat unclear and has therefore been changed to:

"As CRE is the sum of SCE and LCE, annual average values in Table 4 are more

influenced by SCE than LCE, and result in DAV and PAY having the lowest values at ∼ -40 W m-2".
* * *
Referee comment: p. 8, line 4: Looks like 0.9-3.1 to me.

Answer: Thank you. The range from a previous manuscript version had not been updated. The range has now been changed to 0.9 – 3.1.
* * *
Referee comment: p. 8, line 5: Over time or space?

Answer: Rather than add more detail to the sentence, we would prefer to change the sentence to:

". . . SCE and LCE trends both range from positive to negative values".
* * *
Referee comment: p. 8, line 6: The trends for CRE are also mostly insignificant.

Answer: This sentence has been removed in the new version.
* * *
Referee comment: p. 8, line 10: What does "this" refer to?

Answer: "This is most likely the case . . ." has been changed to "The latter is also more likely the case . . .".
* * *
Referee comment: p. 8, line 13: See prior comment about "increasing" and "decreasing" trends. Does "trends" refer to both DSR and DLR here?

Answer: In this case, we do mean an increase or decrease in the trend as we are
discussing a change in fractional cloud cover or cloud type.

The word "trends" refers to both DSR and DLR, but we have included the word twice as the sentence is rather long.
* * *
Referee comment: p. 8, line 18: Is the Sanchez-Lorenzo study relevant to your results? You say it only showed cloud cover trends in the 1970s and 1980s.

Answer: The original sentence: "Sanchez-Lorenzo et al. (2017) reported a decrease in observed and simulated cloud cover during the first two decades of the 1971 – 2005 period over the Mediterranean region which was followed by a subsequent tailing off of the trend".

In this sentence, the word "reported" may suggest that only the 1971 – 1980 period was reported but Sanchez-Lorenzo et al (2017) in fact reported the whole 1971 – 2005 period. As it was not our intention to convey this, the sentence has been changed to:

"Sanchez-Lorenzo et al. (2017) reported the observed and simulated cloud cover for the 1971 – 2005 period, and found negative trends during the first two decades over the Mediterranean region, followed by a subsequent tailing off."
* * *
Referee comment: p. 8, line 19: Change to "included".

Answer: We feel that the term "also covered" is appropriate as we are talking about spatial dimensions. However, we have changed it to "included".
* * *
Referee comment: p. 8, lines 21-23: Another paper that ties changes in DSR to changes in clouds (rather than aerosols) is Parding et al., 2016 (J. Climate).

Answer: Thank you for pointing out this interesting paper, which we completely missed.

An extra paragraph has been inserted at the end of the original paragraph.

"These aspects were more closely investigated with respect to possible changes in synoptic weather patterns by Parding et al. (2016). They observed that an increase in cyclonic and decrease in anticyclonic weather patterns occurred over northern Europe, and contributed to dimming in the 1960s to 1990s based on observational data from the Global Energy Balance Archive (GEBA; Gilgen and Ohmura, 1999)."
* * *
Referee comment: p. 8, line 23: This paper is not listed in the references.

Answer: Thank you. Our mistake. The reference was in the list but it was not in correct alphabetical order. This has now been rectified.
* * *
Referee comment: p. 8, line 25: This would be easier to understand if you reminded us of what the "anomalies" are and what we could expect to learn from them (based on variable dependencies in the parameterization), before providing the numerical results instead of afterwards. That is, remind us that measured values of IWV and T are used for the clear-sky estimates. You might also mention that you mean the Prata param. here, just to be clear.

Answer: The paragraph has been restructured to:

"Through analysis of the trends in the longwave discrepancy, it is possible to assess the strength of radiative forcing components other than due to changes in T2m and IWV. Only these latter two parameters are used in the Prata parameterisation to estimate DLRsim cloud-free. Trend analyses are shown for each station in Table 6. The LLS DAV trend of 3.4 W m$-$2/decade represents 70 % of the overall DLR trend of 4.8 W m$-$2/decade from Table 3. A similarly high value is also found at JFJ, and suggests that 70 % of the overall cloud-free trends at these stations are due to factors other than T2m and IWV."

Referee comment: p. 8, line 29: Which trend, in the anomaly or the DLR?

Answer: The entire paragraph discusses the trends in the longwave anomalies. We therefore feel that adding extra text, here and elsewhere, is uneccessary. However, we have changed

" ...the trend at LOC ..."

to

" ... the LLS trend at LOC ..."

to make the sentence clearer.
* * *
Referee comment: p. 8, lines 33-37: By "anomaly," do you still mean the difference between estimated and measured clear-sky DLR? If so, how do the other authors estimate DLR? Do they include aerosols and trace gases, as mentioned next? It's not clear how results with respect to these variables were obtained.

Answer: The original sentence has been changed from:

"Previous studies (Philipona et al., 2005; Wacker et al., 2011) have investigated the trends in the longwave anomaly but changes in atmospheric gases or aerosol concentrations were not considered to be the cause."

to:

"Previous studies (Philipona et al., 2005; Wacker et al., 2011) have investigated the trends in the longwave discrepancy using similar methods to those in this study. Possible changes in atmospheric gases or aerosol concentrations were investigated but not considered to substantially contribute to the discrepancy."

Referee comment: p. 9, line 8: Blue?

Answer: Thank you. "Blue" has been inserted.
* * *
Referee comment: p. 9, line 18: I assume you mean sky-camera based FCC here?
Referee comment: p. 9, line 18: How do you apply a sky camera method to a pa-
rameterization? Referee comment: p. 9, lines 17-21: This text needs to be clarified.
It sounds like you are trying to evaluate the sky-camera method of estimating cloud
cover, but the results are given in W/m2. What are you actually doing? And why do
you believe that the results are "likely" to improve when more data is available?

Answer: The original sentence was:

"A promising alternative to APCADA to determine the degree of cloud cover is the use
of sky cameras. As only FCC time series at DAV and PAY from sky-camera data were
available for the 2013 – 2015 period, it was tried on the above DLR-IWV parameterisa-
tion. These values are higher than with APCADA (10.0 and 10.1 W m-2) but are likely
to improve (i.e. decrease) when longer time series become available in the future".

This has been changed to the following, and hopefully answers the three comments
from the Referee:

"A promising alternative to APCADA to determine the degree of cloud cover is the use
of sky cameras. However, FCC time series at DAV and PAY from sky-camera data
are only available as of 2013, and hence cannot be used to replace APCADA in this
study based on the 1996 – 2015 period. Instead, the 2013 – 2015 FCC time series
was tested with the above DLR-IWV parameterisation. Rmse values of 13.7 W m-2
and 12.8 W m-2 for all-sky (R2 = 0.80) and cloud-free conditions (R2 = 0.85) were
obtained, respectively. These rmse values are higher than with APCADA (10.0 and
10.1 W m-2) but are likely to improve (i.e., decrease) when longer FCC time series
from sky cameras become available in the future."

Referee comment: p. 9, line 35: This contradicts text on page 6.

Answer: Thank you. The text has been changed to:

"All-sky and cloud-free DSR trends are in the ranges 0.6 – 4.3 W m-2/decade and 3.1 – 3.3 W m-2/ decade, respectively. Half of the trends are significant at the 90% confidence level."

The text in the main body of the manuscript (originally p.6 line 23) has also been updated to:

Trends in all-sky DSR are in the 0.6 – 4.3 W m-2/decade range with a significance at the 90% confidence level except for DAV. Cloud-free trends for DAV and LOC are similar (3.1 and 3.3 W m-2/ decade, respectively) but are noticeably different for PAY and JFJ (10.6 and -9.5 W m-2/decade, respectively).
* * *
Referee comment: p. 9, line 38: Estimated, because the clear-sky values don't come from measurements.

Answer: "Estimated" inserted.
* * *
Referee comment: p. 10, line 1: Since this is a big range, you might want to say where it is high and low or just that it varies by location.

Answer: ". . ., depending on location, . . ." inserted.
* * *
Referee comment: p. 10, lines 9-10: It would be useful to compare the magnitudes of the detected trends and measurement accuracy in the text. Otherwise we are left with the impression that the standard deviations given in the tables accurately represent

your confidence in the results.

Answer: This question could be tackled in several different ways but after studying the literature, we decided to frame our answer in terms of the 95% confidence interval of the trends. These have been added to Tables 3 and 5 but only for the LLS method for clarity. We could have calculated the range in the trend by adding and subtracting the measurement uncertainty from each data point in the time series but this range would in fact be smaller than the 95% confidence interval.

The following sentence was added to Section 3.2.:

"The 95 % confidence intervals of the DLR trends, as well as those for meteorological and DSR trends, are shown in Table 3. Interval values are relatively low in all cases, and is in large part due to the long time series. If the instrumental uncertainties are taken into account by the trend analysis, then 95 % confidence intervals are unchanged to two decimal places. However, our main reason to have confidence in trend results, rests on whether they are significant or not at the 95 % confidence level, which has been demonstrated in Table 3".

The following sentence was added to Section 3.3.1:

"However, it should be noted that no trends are significant at the 95 % confidence level with only PAY significant at the 90 % level. Although the absence of any significant trend hampers further reliable interpretation, it is nevertheless interesting to consider what results in Table 5 suggest."

The following sentences in the Abstract have been augmented with the text in bold:

"The trends of meteorological parameters and surface downward shortwave and long-wave radiation (DSR, DLR) were analysed at four stations (between 370 and 3580 m asl) in Switzerland for the 1996 – 2015 period. Ground temperature, specific humidity and atmospheric integrated water vapour (IWV) increased during all-sky and cloud-free conditions. All-sky DSR and DLR trends were in the ranges 0.6 – 4.3 W m-2/decade

and 0.9 – 4.3 W m-2/decade, respectively, while corresponding cloud-free trends were -2.9 – 3.3 W m-2/decade and 2.9 – 5.4 W m-2/decade. Most trends were significant at the 90 % and 95 % confidence levels. The cloud radiative effect (CRE) was determined using radiative transfer calculations for cloud-free DSR and an empirical scheme for cloud-free DLR. CRE decreased in magnitude by 0.9 – 3.1 W m-2/decade (only one trend significant at 90 % confidence level), which implies a change in macrophysical and/or microphysical cloud properties. Between 10 and 70 % of the increase in DLR is explained by factors other than ground temperature and IWV. A more detailed, long-term quantification of cloud changes is crucial and will be possible in the future as cloud cameras have been measuring reliably at two of the four stations since 2013."

The following sentence in the Conclusions has been augmented with the text in bold:

"The estimated net radiative cooling due to clouds, the CRE, decreased in magnitude by 0.9 – 3.1 W m-2/decade over the 1996 – 2015 period, although no trends were significant at the 95% confidence level."
* * *
Referee comment: p. 10, line 15: Should be ", e.g., cloud type,"

Answer: Changed.
* * *
Referee comment: p. 10, line 18: Change to "that".

Answer: As a native British English speaker, I would argue that our version is correct. As our ACP paper will undoubtedly be language-corrected before publication, we would recommend that ACP decide.
* * *
Referee comment: p. 10, line 26: Which author, and what about the others? All sources of funding should be recognized.

Answer: In our case the "author" is the "main author". We've never been asked to
include the funding sources of other authors. However, it may actually not be necessary
as all other authors have full-time positions at their respective institutes involving no
third-party funding. However, we will change the text to whichever format ACP requires.
* * *
Referee comment: Table 2: Do I understand correctly that these values have been
published previously?

Answer: Results for the period 1996 – 2007 were reported by Wacker et al (2011). We
state this in the original manuscript on page 5 line 28. Table 2 refers to updated values
for the 1996 – 2015 period. Hence, this data has not been published before.
* * *
Referee comment: Table 3: Any idea why the clear-sky DSR decreases at JFJ but not
at the other stations? Is there a reason cloud cover trends aren't included in this table?

Answer:

We would prefer not to speculate why a there is an overall negative trend in DSR at
JFJ for the 1996 – 2015 period. In the original manuscript on p.6 lines 23-29, we
discuss the large negative and positive trends at JFJ and PAY. We also point out in the
footnotes of Table 3 that the trends are less negative and positive over different time
periods. Without going into further speculative detail we have therefore changed the
following sentence:

"In both cases, homogeneity analysis (described further below) does not suggest that
a stepwise change occurred due to a change in instruments etc., so whether these
trends continue into the future will have to be further monitored."

to:

"Only the Pettitt homogeneity test suggested that a discontinuity in the trend occurred

at PAY and JFJ (both, $p < 0.05$). No discontinuities were found in the DAV and LOC DSR trends. At present, the reason(s) for these cloud-free trends at PAY and JFJ for 1996 – 2015 are unknown and will have to be further monitored. The SCE, LCE and CRE are not affected by these results as they are calculated with all-sky data."

We have not included cloud-cover trends as the method to determine cloud-cover is based on a parameterisation (APCADA). It is therefore our opinion that a discussion of parameterised cloud-cover trends would not add further insight to the scientific discussion. The 6-year data-set of sky-camera measurements is unfortunately too short at present.
* * *
Referee comment: Table 3: The trend is the slope, i.e., unit/decade, not the slope/decade (which would be the change in the slope per decade).

Answer: Thank you for highlighting this error. This has been changed to unit/decade.
* * *
Referee comment: Table 4: The text on page 7 lists fairly large biases and RMSEs for the SW clear-sky fluxes, as much as 17% of the means and 3x the standard deviations of SCE shown in this table, respectively. Are these errors important to the SCE estimates?

Answer: After having re-analysed data used in this section, we discovered that the values for both the DSR and DLR discrepancies that we gave were from an earlier incorrect version of the paper. The correct values have now been inserted, which are similar to the instrumental uncertainty. The uncertainty estimates of SCE, LCE and CRE are therefore "correct". The revised text has been moved to Section 2.3, and is:

"Validation of the cloud-free models was accomplished by determining the shortwave and longwave discrepancies (observed cloud-free fluxes – simulated cloud-free fluxes). The mean bias and rmse of the shortwave discrepancies were <3.5% and <8.5%

(Wacker et al., 2013), respectively, and $\sim$ -0.1 W m-2 and $\sim$3.9 W m-2 for the longwave discrepancies at all four stations. The mean biases are thus similar to the measurement uncertainty of the respective radiometers (Wacker et al., 2013)."
* * *
Referee comment: Figure 1: If you have no comments about the trends determined using the Weatherhead method, why are they included?

Answer: The Weatherhead method is in fact the linear least squares (LLS) method which is used throughout the manuscript, and is first described in section 2.4.

We have therefore changed the original caption text from:

"Each panel also shows trend results from linear least squares analysis using the Weatherhead et al. (1998) method."

to the following for clarity:

"Each panel shows trend results from linear least squares analysis in Table 3."
* * *
Referee comment: Figure 3: Did you also check the AOD data for artificial jumps? There looks like there might be a discontinuity in the PAY data around 2011.

Answer: Thank you for noticing this. In fact, we visualised the wrong data series for the 2013-2015 period in Figure 3. It should have been a climatology, as mentioned in Section 2.3, and not the data shown. Both LOC and PAY AOD time series for the 1996 – 2012 period were homogeneous (p > 0.05). Figure 3 has therefore been updated.

Despite this small problem, the correct data were used for the radiative transfer calculations, and therefore no SCE or CRE results need to be changed.
* * *
Referee comments: Clarity of presentation:

At some points, additional detail or improved clarity is needed. Questions about the meaning of certain phrases and suggestions for wording changes are included in the accompanying PDF file.

Whenever possible Note: A comma is required - before (and after) a phrase starting with "which" - after (and before, if they're not inside parentheses) "e.g." or "i.e." - before "etc."

Answer: Thank you. We thank the referee for reviewing our paper in such detail. We have endeavoured to correct the paper according to his/her recommendations.

---

## Author Comment (AC2) · 20 Jul 2019

20 July 2019

Dear Editor and Referees.

Please find our corrections below. We thank both Referees for their thoughtful comments and detailed corrections. It has taken longer than anticipated to correct our paper, but the effort has definitely been worthwhile. We therefore hope that we have answered the questions as best as possible and that the latest version meets with the Referees approval.

During the revision, we came across an interesting review paper by Pepin et al. (2015), published in Nature Climate Change. Two small paragraphs have been added to the new paper on page 7 line 21 and page 9 line 41.

Pepin, N., Bradley, R. S., Diaz, H. F., Baraër, M., Caceres, E. B., Forsythe, N., Fowler, H., Greenwood, G., Hashmi, M. Z., Liu, X. D. and Miller, J. R.: Elevation-dependent warming in mountain regions of the world, Nature Climate Change, 5, 424, doi: 10.1038/nclimate2563, 2015.

Best Regards Stephan Nyeki

PMOD/WRC Davos, Switzerland

  Referee 2: Comments

General comments

It is important to investigate the trends of radiation and cloud radiative effect at the surface and put forward the possible explanations that account for the phenomenon. However, there are some weakness that requires more supporting material. The clouds identification methods are not accurate which may induce a contamination of radiation fluxes, since a slight change of cloud cover may significantly influence the radiation fluxes at the surface. Aerosol burden also has significant effects on shortwave radiation at the surface. Using climatological average of AOD to fill in the missing data of AOD during 2013 through 2015 may cause an artificial error in the trend analysis. Without any detailed description of how clouds change, it is quite arbitrary and blurry to infer the relationship between the variations of CRE and clouds.

Question: The clouds identification methods are not accurate which may induce a contamination of radiation fluxes, since a slight change of cloud cover may significantly influence the radiation fluxes at the surface.

Answer: This is correct. However, we would point out that sky-camera measurements have only been conducted for several years, and reliable data are only available for PAY

and DAV. The only way to enable cloud-free DSR and DLR analyses to be extended back to the 1990s is with a proxy parameterisation of cloud-cover, APCADA in our case. The original manuscript already discusses the advantages and disadvantages of using APCADA. In our opinion, this is currently the only way to conduct such analyses. However, we have made some changes to Section 2.2 so that the advantages of using sky-cameras are better highlighted.

New text: "Apart from these aspects, the use of proxy parameterisations for cloud cover will introduce uncertainties, but we estimate that these are generally low. A more accurate assessment will only be possible when cloud cover data from sky cameras are long enough to conduct reliable time series analysis, which is generally a period of 10 years and longer. While cloud cover can be accurately and objectively determined with sky cameras, measurements are only available during daylight hours."
* * *
Question: Aerosol burden also has significant effects on shortwave radiation at the surface. Using climatological average of AOD to fill in the missing data of AOD during 2013 through 2015 may cause an artificial error in the trend analysis.

Answer: We agree. On the other hand, the 17-year AOD time series is only being extended with a 3-year climatology to 20 years, so a large deviation from the prevailing trend is not to be expected. We have therefore introduced the following sentence:

"While this may introduce an error in the AOD trend, a large change is not expected in the 18-year AOD time series."
* * *
Question: Without any detailed description of how clouds change, it is quite arbitrary and blurry to infer the relationship between the variations of CRE and clouds.

Answer: We agree with the Reviewer. We have therefore changed our discussion at several points in the text so that a change in macro and microphysical cloud properties

is mentioned rather than a change in cloud cover or in cloud type. Please see further comments below (Referee comment: 6. Page 8 and line 15).
* * *
Specific comments

Referee comment: 1. Page 5 and line 25: 'In contrast, the specific humidity and IWV are higher during all-sky conditions which in turn results in higher DLR values.', please mention the source of the specific humidity data?

Answer: Specific humidity was calculated from T2m, RH and pressure. This has been included in a revision of the paragraph on the IWV parameterisation. Please see next Referee comment.
* * *
Referee comment: 2. Page 5 and line 30: 'IWV at JFJ was based on a widely-used parameterization using T2m and relative humidity', where is relative humidity data from and what is the accuracy of this parameterization? Please add description of these in the section of methods and data.

Answer: This small paragraph has been moved to Section 2.1 and has been changed to:

"As a result, IWV at JFJ was based on a commonly-used parameterisation by Leckner (1978) using T2m and RH. Gubler et al. (2012) estimated that the uncertainty in IWV using this parameterisation was up to 100 %."

Gubler, S, Gruber, S., and Purves, R. S.: Uncertainties of parameterized surface downward clear-sky shortwave and all-sky longwave radiation, Atmos. Chem. Phys., 12, 5077-5098, doi:10.5194/acps-12-5077-2012, 2012.
* * *
Referee comment: 3. Page 5 and line 35: 'which are probably associated with synoptic scale weather patterns', what kind of weather patterns are they?

Answer: Rather than go into further detail on this subject for which there are few observational studies, we have removed the latter part of the sentence so that the following remains.

"Weak seasonal variations are seen to occur at all sites".
* * *
Referee comment: 4. Page 6 and line 25: 'the DSR trend at PAY is not monotonic but steeply', as DSR has obvious changes, could you show its trend at four sites like Figure 1?

Answer: We would prefer not to add another figure with, in our view, little scientific value to the overall paper. The word "steeply" was an overemphasis and has therefore been removed.
* * *
Referee comment: 5. Page 7 and line 20: what are the '<' and '-<'?

Answer: Thank you for highlighting this. This has been corrected to <-0.5 Wm-2 and <4 Wm-2.
* * *
Referee comment: 6. Page 8 and line 15: 'suggesting that a decrease in fractional cloud cover or a different cloud type has occurred during the 1996 – 2015 period', there are many factors may affect cloud radiative effects such as cloud height and optical depth. A decrease of CRE magnitude doesn't mean there can be the decrease of cloud fraction. What are the variations of different clouds during the 1996 – 2015 period?

Answer: We agree with the Reviewer that both, macrophysical cloud properties (e.g., cloud cover, cloud base height (cloud base temperature), cloud top height etc.) and microphysical cloud properties (e.g., cloud optical thickness, cloud droplet size, cloud particle size distribution, liquid water content, liquid water path, ice water content, hydrometeor size, hydrometeor size distribution, hydrometeor phase etc.) determine CRE and thus changes in CRE are a result of changes in these parameters. Regarding cloud fraction, we refer to Fig. 3 and Fig. 4 in Aebi et al., 2017 which indicate an increase in the magnitude of cloud radiative effects with increasing cloud fraction, particularly in the long-wave. In addition, various studies (listed in lines 16-23) conclude that cloud cover over Europe has decreased and thus it is likely that this decrease has contributed to the decrease of CRE.

Due to the lack of long-term cloud observations (macrophysical and microphysical cloud properties) it is not possible to determine the variations of different clouds during the 1996 – 2015 period. Indeed, continuous active remote sensing techniques are only available at Payerne but time series are not longer than 10 years. Cloud type observations from human observers, which are subjective to some extent and difficult to analyse, were finally stopped in 2005.

The original sentence on p.1 lines 18-19 has been changed from:

"CRE decreased in magnitude by 0.9 – 3.1 W m-2/decade which implies a reduction in cloud cover and/or a change towards a different cloud type over the four Swiss sites."

to:

"CRE decreased in magnitude by 0.9 – 3.1 W m-2/decade which implies a change in macrophysical and/or microphysical cloud properties."

The original sentence on p.8 lines 9-10 has been changed from:

"As a result of the positive CRE trends in Table 5, there is an overall decrease in the CRE magnitude, suggesting that a decrease in fractional cloud cover or a change

towards a different cloud type has occurred during the 1996 – 2015 period.2

to:

"As a result of the positive CRE trends in Table 5, there is an overall decrease in the CRE magnitude, suggesting that changes in macrophysical and/or microphysical cloud properties, which determine CRE, have occurred during the 1996 – 2015 period."

The original sentence on p.8 line 16 has been changed from:

"A reduction in cloud cover over Europe . . .".

to: A decrease in cloud cover might be one of the cloud parameters which has contributed to the decrease of the CRE magnitude. Indeed, a reduction in cloud cover over Europe. . ..

The original sentence on p.8 line 24 has had the following text added on:

"Apart from changes in cloud cover and other macrophysical cloud properties, microphysical cloud properties can also have a substantial impact on CRE. However, the observation of these properties using active remote sensing techniques is limited to a few super sites-worldwide while long-term time series are rarely available. The same is also valid at the four Swiss SACRaM stations in this study. Macrophysical and microphysical cloud properties have only been routinely measured at PAY since 2010 and 2005, respectively. In addition, cloud observations from human observers were discontinued in 2000 – 2005 at the SACRaM locations."

The original sentence on p.9 line 39 has been changed from:

". . . over the 1996 – 2015 period, which implies a decrease in cloud cover or a change towards a different cloud type."

to:

"...over the 1996 – 2015 period, although no trends were significant at the 95% confidence level. This decrease in CRE is probably caused by variations in macrophysical and microphysical cloud properties. However, it is not possible to determine and quantify, which cloud properties have changed and contributed to the decrease in CRE due to the lack of corresponding continuous long-term observations."